


# Multisectoral analysis of drought impacts and management responses to the 2008-2015 record drought in the Colorado Basin, Texas: A blueprint for regional multisectoral drought impact assessment

Stephen B. Ferencz[1], Ning Sun[1], Sean Turner[1], Brian A, Smith[2], Jennie S. Rice[1]

[1]Pacific Northwest National Laboratory, Richland, WA, United States
[2]Barton Springs Edwards Aquifer Conservation District, Austin, TX, United States

*Correspondence to*: Stephen B. Ferencz (stephen.ferencz@pnnl.gov)

**Abstract.** Drought has long posed an existential threat to society. Engineering and technological advancements have enabled the development of complex, interconnected water supply systems that buffer societies from the impacts of drought, enabling growth and prosperity. However, increasing water demand from population growth and economic development, combined with more extreme and prolonged droughts due to climate change, pose significant challenges for governments in the 21st century. Improved understanding of the multisectoral impacts and adaptive responses resulting from extreme drought can aid in adaptive planning and highlight key processes in modelling drought impacts. The record drought spanning 2008 – 2015 in the Colorado Basin in the state of Texas, United States serves as an outstanding illustration to assess multisectoral impacts and responses to severe, multi-year drought. The basin faces similar water security challenges as across the Western U.S., such as: groundwater depletion and sustainability, resource competition between agriculture and growing urban populations, limited options for additional reservoir expansion, and the heightened risk of more severe and frequent droughts due to climate change. By analysing rich, high-quality data sourced from nine different local, state, and federal sources, we demonstrate that characterizing regional multisector dynamics is crucial to predicting and understanding future vulnerability and possible approaches to reduce impacts to human and natural systems in the face of extreme drought conditions. This review reveals that, despite the severe hydrometeorological conditions of the drought, the region's advanced economy and existing water infrastructure effectively mitigated economic and societal impacts.

## 1. Introduction

Droughts threaten modern civilizations in a variety of ways (van Dijk et al. 2013, Wilhite et al. 2007). Prolonged dry spells cause depletion of terrestrial water resources, leading to water use restrictions and shortage (Lund, et al. 2018), reduced crop yields and loss of pasture (Gupta et al., 2020; Kuwayama et al., 2019), impaired electricity generation from hydroelectric and thermoelectric facilities (van Vliet et al., 2016; Voisin et al., 2020), degradation of water quality (Ahmadi and Moradkhani, 2019), forest loss through tree mortality (Brodribb et al., 2020) and forest fire (Littell, et al. 2016), and reduced primary productivity of vegetation (Stocker et al., 2019; Xu et al., 2019). These impacts spawn a myriad of second-order effects. For





instance, loss of water-dependent electricity generation can reduce the reliability of the power grid (Turner et al., 2021) or shift generation onto resources that cost more to run or emit more carbon (O'Connell et al., 2019). In some cases, the impacts of a local drought can carry national or global implications, such as by increasing crop prices and altering global food trade

networks (Lal et al., 2012; Marston and Konar, 2017).

The need to understand possible impacts from drought is underscored by anticipated intensification of drought in some world regions in the 21st century due to climate change (Cayan et al., 2010; Cook et al., 2018; Trenberth et al., 2014), manifesting large reductions in surface water availability over large portions of the globe (Schewe et al., 2014). In some regions, climate change has already increased the joint probability of hot and dry conditions that produce more severe drought impacts

(Sarhadi et al., 2018).

There is no single quantitative definition of drought (Kuwayama et al., 2018). Drought can be defined by many metrics of water deficit, such as reduced precipitation (meteorological drought) often combined with increased potential evapotranspiration, soil moisture deficit impacting vegetation (soil moisture drought or agricultural drought), reduced surface water flows and groundwater levels (hydrological drought), and reduced reservoir storage (reservoir drought) (Van

Loon et al., 2015). The intensity and duration of meteorological drought influences the severity of other types of droughts; for example, a short, intense meteorological drought can result in a severe agricultural drought. The impacts of meteorological drought can also be exacerbated by human actions (Van Loon et al. 2016), such as increased diversions from streams resulting in more severe hydrological drought (reduced streamflow) or withdrawals from reservoirs initiating or exacerbating reservoir drought.

Since extreme drought is rare (by definition), there are a limited number of 21st century case studies available to document and synthesize its impacts. Examining each case is essential to better understanding the complex dynamics of drought propagation, the resulting multisector impacts and responses to drought in modern society, and critical lessons learned to better prepare for future droughts. The aim of this paper is to provide such a case study through a detailed examination of the 2008-2015 drought in the Colorado Basin, TX. The paper is organized into the following sections: background on the

drought of record, e.g., basin's hydroclimate, water supply, and sectoral water use and overview of data sources (Section 2); analysis of multisectoral impacts during the 2008-2015 drought of record (Section 3); changes to water planning, policy, and management following the drought of record (Section 4); and finally, a discussion of key challenges facing the basin, potential pathways to a more resilient water future, and a comparison to economic impacts of recent droughts in other advanced economies (Section 5).

**2. Background and Data**

**2.1 Basin Geography and Sectoral Water Use**


The Colorado Basin, Texas (Figure 1e) is facing significant municipal-agricultural-energy water nexus challenges, offers a compelling case study for multi-sectoral drought impact analysis. The basin is divided into three water management regions (Figure 1a), marked by diverse hydroclimate and distinct differences in water use, reliance on surface water versus

groundwater, and sectoral water demand (Table 1). Here, water use refers to total withdrawals, not consumptive use.

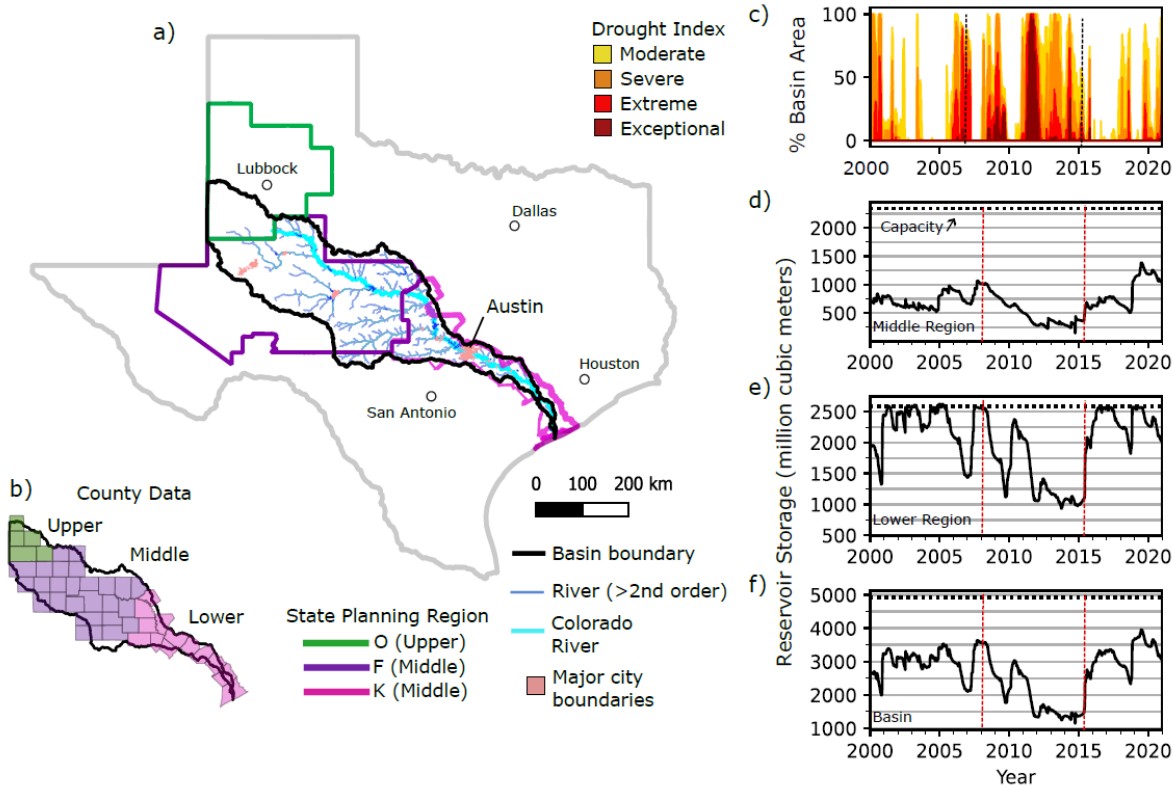

**Figure 1**: The Colorado Basin (a) spans 800 km across the central part of Texas and has a drainage area of 102,000 km$^2$. Its headwaters are in the arid north-western part of the state, and it flows southeast the Gulf of Mexico along the Texas coast. The basin spans three state water planning regions: Region O (upper), Region F (middle), and Region K (lower). All regional

data presented is for counties within the basin footprint (b). U.S. Drought Monitor Drought Index categories for the basin from 2000 to 2020 (c). Reservoir storage for the middle region (d), lower region (e), and total basin (f) in million m$^3$.

The basin's hydrology is characterized by highly variable seasonal streamflow prone to multi-year drought periods (Wurbs, 2021). There is a markedly increasing precipitation gradient from the western upper region to the easter lower region, which

greatly influences surface water availability and the ratio of surface water to groundwater use across the basin (Table 1). The upper region is almost entirely dependent on groundwater sourced from the Southern High Plains Aquifer. In contrast, the lower region receives more than half its annual supply from surface water. Lower region reservoirs are the critical supply for



accommodating the city of Austin's municipal demands and irrigators. The middle region is heavily reliant on groundwater for agriculture but uses surface water to meet 60-70% of its municipal demand. Overall, the middle region uses less than 20% of the surface water of the lower region.

| | | Average Water Use | | | Average Sectoral Water Use | | | | Reservoir | |
|---|---|---|---|---|---|---|---|---|---|---|
| Region | Population | Total | SW | GW | Agriculture | Municipal | Industrial | Thermo-electric | Average Storage | Capacity |
| Lower | 1,390,569 | 1,142 | 850 | 292 | 719 | 283 | 55 | 85 | 2,255 | 2,632 |
| Middle | 536,774 | 437 | 141 | 296 | 292 | 128 | 15 | 3 | 719 | 2,337 |
| Upper | 52,204 | 1,191 | 6 | 1,185 | 1,176 | 13 | 2 | 0 | na | na |

**Table 1:** Summary of regional water use, population, and reservoir storage. Annual average water use, sectoral water use, and reservoir storage volumes in $10^6$ m$^3$ (data for 2000 – 2007, pre-drought period) for the three planning regions. Only includes counties shown in Figure 1b.

Water use and population are unequally distributed amongst the three regions (Table 1). The sparsely populated, heavily agricultural upper region and densely populated lower region both use more than twice the water of the middle region. Prior to the drought (2000-2007), the agriculture sector was the largest water user in all three regions, accounting for 99% of all water use in the upper region and between 50 and 70%, in the middle and lower regions. Municipal use was the second largest sector, representing 25-30% of annual water use in both the lower and middle regions. Industrial and thermoelectric use was less significant in all three regions, accounting for 3-7% of annual use.

**2.2 The 2008 – 2015 Drought of Record**

The 2008-2015 drought is officially recognized as the drought of record for two of the three planning regions (lower and middle regions, Figure 1) in the basin (Texas Water Development Board (TWDB), 2022a). This drought period is characterized by a combination of reservoir and meteorological drought, spanning the time between lower basin reservoirs resetting (Figure 1b) and the end of widespread meteorological drought conditions (Figure 1a). The drought consisted of two dry periods (2008-2009 and late 2010-2015) separated by a relatively wet year in 2010 (Figure 1a). Before 2008-2015, the region's most severe drought on record took place in the 1950s (TWDB, 2022a). Five key factors that make the two droughts different are a combination of climate (natural) and human system factors:

(1) Rapid onset of extreme drought. A record low state-wide Palmer Drought Severity Index (PDSI) of -8.06 occurred just 14 months into the 2011-2015 period whereas the drought of the 1950s took 72 months to reach a record low PDSI of -7.77 (TWDB, 2017).





(2) Record meteorological drought combined with prolonged record heatwaves in 2009 and 2011 (hot-dry drought), and June, July, and August 2011 were 1.4 °C higher than the next hottest summer on record (Neilson-Gammon, 2012).

(3) Sustained, multi-year record low reservoir storage in the basin from 2012-2015 (persistent reservoir drought).

(4) Three times larger basin population with 80% of the population increase in the heavily surface water reliant lower region.

(5) In the 1950's the basin was a largely agrarian economy, in contrast with the predominantly urban, industrialized economy in the 21st century (TWDB, 2022b).

## 2.3 Data

Data was obtained from a diverse array of publicly available sources to understand and characterize the breadth of multisectoral impacts (Section 3) and management responses (Section 4) in the basin (Table 2). Much of the data was available at the annual temporal resolution at the county scale. For these cases, we primarily aggregated the county-level data for each of the three planning regions. Some of the data categories contained an overabundance of records, either hundreds or thousands of locations with hydrological time series data (streamflow, water quality) or numerous metrics associated with annual, county-level data (GPD, employment, crop). To determine region-specific drought impacts, we referenced region-specific literature and regional planning documents to inform most relevant locations and metrics. Data on water supply planning and management were primarily sourced from regional water plans for the three water planning regions, and were supplemented by municipal and utility planning reports, where appropriate. A unique aspect of this study is the extensive review and analysis of grey literature related to water planning. Our characterization of impacts and planning responses is informed by reviewing thousands of pages of planning documents and reports. Our analysis was also informed by interviews with subject matter experts who have experience in city, regional, state, and utility-scale water planning. Data on county-level water supply projects was assembled from each of the regional water plans into a database with supply type, unit cost, supply volume, and sector. Costs were converted to 2022 values using the annual consumer price index for time series data on sectoral GDP (3.5) and water supply unit cost analysis (4.2).

| Data Category | Description | Source/Agency |
|---|---|---|
| Water use | Annual sectoral SW and GW volumes by county (2000 -2020) | TWDB, 2023 |
| Reservoir Storage | Daily reservoir storage (1940 - 2021) | TWDB, 2022c |
| Streamflow | Daily gauged streamflow (2000 - 2020) | USGS, 2023 |
| Water quality | Field water quality samples at river and lakes monitoring locations (2000 - 2020) | TCEQ, 2023 |
| Crop | Annual crop production and harvested area by county (2000 - 2020) | USDA, 2023 |



| Cattle | Annual cattle herd size by county (2000 - 2020) | USDA, 2023 |
|---|---|---|
| Population | Decadal estimates (1940 - 2020), annual estimates (2001 - 2020) by county | US Census, 2022 TWDB, 2022c |
| Wildfire | Annual acres burned by county (2008 - 2015), acres burned state-wide (2002 - 2021) | NOAA, 2022 |
| Gross Domestic Product (GDP) | Annual sectoral GPD by county (2000 - 2020) | BEA, 2022 |
| Employment | Annual sectoral employment by county (2000 - 2020) | BEA, 2022 |
| Energy Production | Monthly production by power plant (2001 - 2021) | EIA, 2022 |
| Drought Classification | Weekly drought classification (% area under each drought threshold) for basin (2000 - 2020), weekly drought classification maps (2008 - 2015) | U.S. Drought Monitor, 2023 |
| Well installation by sector | Annual well installations by sector by county (2001 - 2021) | TWDB, 2022d |
| Planned future supply | Recommended water supply projects to meet future sectoral demand. County-level data aggregated for each planning region. (2011, 2016, 2021) | Regional Water Plans* |
| Unit cost by supply type | Unit cost for each recommended water supply project. County-level data aggregated for each planning region (2011, 2016, 2021) | Regional Water Plans* |

**Table 2:** Data sources for multisector impacts and water management response characterization. *Regional water plans include 2010, 2015, and 2020 regional plans for each of the planning regions.

## 3. Analysis of Multisector Dynamics and Impacts

130

A summary of the multisector dynamics of the 2008-2015 drought of record is illustrated in Figure 2 using a directed acyclic graph (DAG). A DAG, also known as an influence diagram, is a compact way to present complex causal relationships pictorially; it can also be implemented mathematically to understand causal inferences (not performed for this study) (Howard and Matheson, 2005; Schachter, 1987). Each node represents a state variable and each arrow shows the direction of
135    influence. Feedback loops are not permitted in influence diagrams—those would need to be shown by connecting the relevant nodes between two influence diagrams across a time step. For our purposes, Figure 2 shows the cascading impacts that stemmed from the initial trigger of severe meteorological drought. The numerous nodes and links convey the highly multisectoral, interconnected nature of drought impacts; most nodes are influenced by multiple upstream states and contribute to multiple downstream outcomes.

140

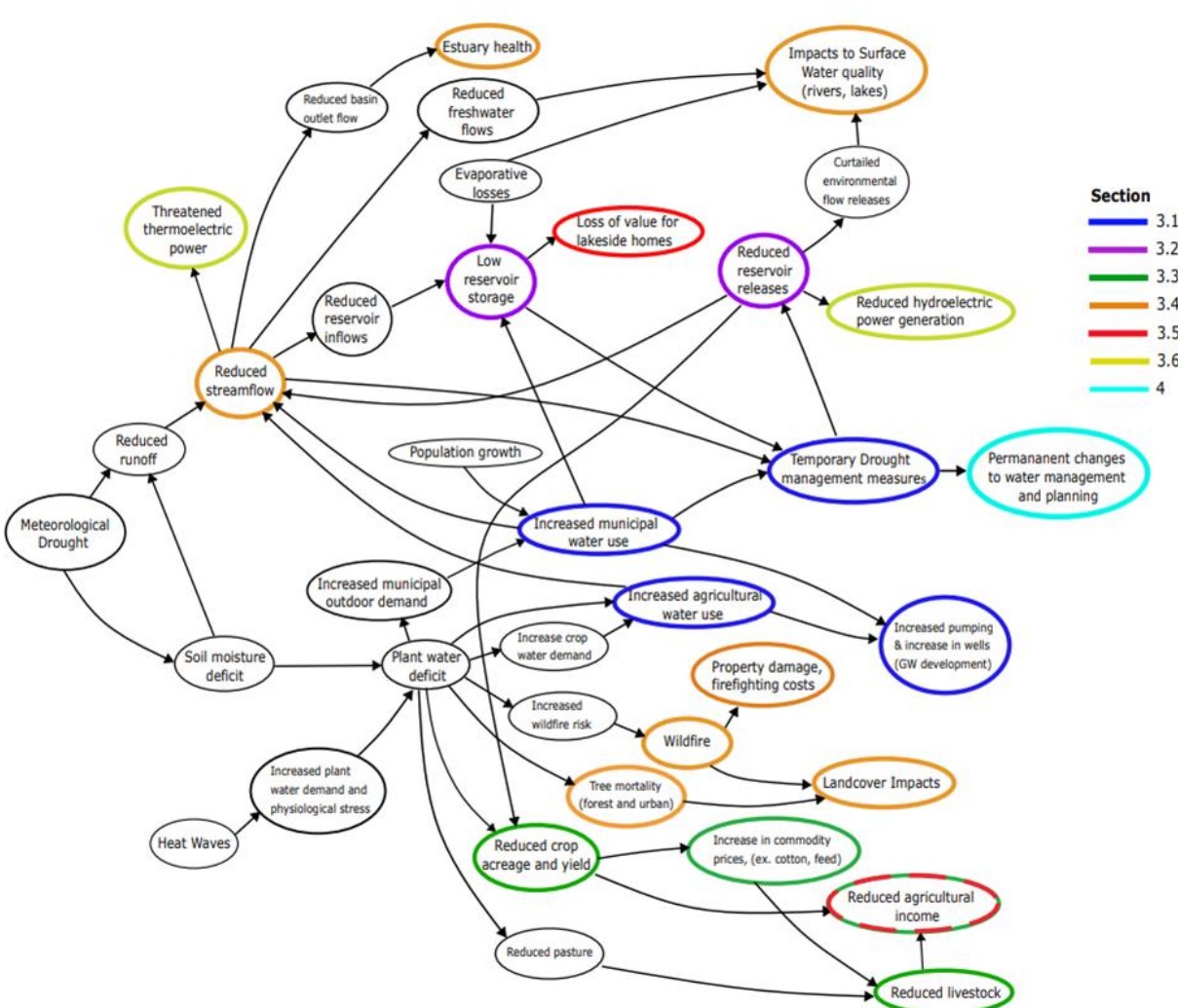

**Figure 2:** Influence diagram describing multisector impacts and interactions during the drought. Arrows depict influence of upstream state variables on a downstream state variable and can be interpreted as connecting causes and effects. Colors indicate multisector impacts covered in each of the corresponding results sections.

The influence diagram also provides an efficient framework to trace downstream outcomes (what resulted from state X?) or upstream causes (what sequence of states led to outcome Y?). The diagram presented here is not intended to be exhaustive but aims to capture key impacts covered in this review. Indeed, many of the individual nodes or drought-categories within





the diagram could be the subject of in-depth studies on their own. The aim of this work is to highlight the variety and causal nature of multisectoral impacts during drought. As a static illustration, Figure 2 does not provide information on the temporal nature (timing, frequency, duration) or severity of impacts. For example, some impacts occurred months into the drought (agriculture in early 2008) while others took years to develop (estuary impacts did not occur until 2011). Some were brief but intense (wildfire) and others were prolonged (reservoir drought from 2011-2015). We note these temporal dynamics in the text.

This section provides context for the causal relationships, temporal characteristics, and severity of drought impacts across multiple sectors in the Colorado Basin. The following subsections provide analysis of impacts to sectoral water use (3.1), reservoir storage (3.2), agriculture (3.3), the environment (3.4), the power sector (3.5), and the economy (3.6), and Figure 2 is color-coded to the endpoint impacts discussed in each subsection.

## 3.1 Sectoral Water Use

As shown in Figure 2, water use data for the basin indicated that meteorological drought impacts propagated to alter sectoral demand (e.g., agriculture, municipal), sectoral water availability (e.g., surface water), and surface water and groundwater use. The onset of the drought in 2008 marked the highest amount of water use in the middle and upper regions (from increased groundwater use), while 2011 was the largest annual water use in the lower region (from both increased surface water and groundwater use) (Figure 3). Notable regional differences in year-to-year variability of water use during the drought were driven primarily by agriculture (Figure 3), while municipal use (second largest sector) showed comparatively little absolute (volumetric) fluctuation when compared to total water use within each region (Figure 3). As the drought progressed, surface water use declined in the middle and lower regions, reflecting reservoir conservation measures and temporary drought management measures enacted by municipal water providers (SI Figure 1). During the last three years of the drought (2013-2015), surface water use in the lower region was 40% less than from 2008-2010, while in the middle region, surface water use decreased by 19%. In contrast, average groundwater use in the middle and lower regions showed little change during the drought with the lower region increasing by 5% and the middle region decreasing by 8%, when comparing groundwater use during 2008-2010 versus 2013-2015.

A sectoral use trend unique to the middle region was rapid growth of industrial use from 2008 to 2020, which increased over 150% between 2008 and 2015 and continued to grow from 2016-2020 (Figure 3h). This growth was almost entirely associated with unconventional (fracking) oil and gas development (Region F, 2020). Oil and gas development often uses non-potable sources, such as saline or brackish groundwater and treated municipal wastewater, so this sectoral use does not have to compete with fresh sources needed by municipalities or agriculture (Region F, 2020). While this large sectoral increase occurred during the drought, it was not influenced by drought and is not considered a drought impact (i.e., not in





Figure 2). Thermoelectric water use in the basin increased by an average of 12.4% during the drought compared to the pre-drought period and two of the highest use years occurred during the drought (2009 and 2012). Although not visually apparent on Figure 3j due to its relatively small magnitude compared to other sectoral water uses, there was a 540% increase in groundwater use for thermoelectric water supply in the lower region following the drought (1.58 million m$^3$/yr from 2008-2013 growing to 10.17 million m$^3$/yr from 2015-2020), reflecting a transition to a more drought-tolerant supply.

**Figure 3:** Population growth (a-c), annual surface water (SW) and groundwater (GW) use (d-f), total sectoral use (g-i), and sectoral GW use (j-l) from 2000 – 2019 in the three planning regions. This data only includes counties shown in Figure 1b.

Comparing annual municipal and agricultural use during and following the drought reveals significant shifts in surface water and groundwater use for the two largest sectors in the basin (Figure 4). Compared to the pre-drought period (2000-2007),





agricultural surface water use during the drought declined by an average of 36% in the lower and 38% middle region, and
these reductions persisted over the 2016-2020 post-drought period (Figure 4a). Following reservoir conservation measures in
2012, which curtailed agricultural supply (Figure 2), lower basin agricultural surface water use was 65-77% less than the
pre-drought period. A consequence of reduced agricultural surface water availability in the lower region was an increase in
groundwater use (Figure 4a) and well installations (SI Figure 2) during the drought and post-drought periods (Figure 2).

Average agricultural groundwater use in the lower region was 33% higher compared to the pre-drought period and in 2011 it
was 84% higher, while in the middle region average use was 21% higher during the drought and 42% higher in 2008.
Increased municipal surface water use in the lower region during and following the drought (Figure 4b) is reflective of the
large population growth in the region, which grew by over 450,000 residents between 2008 - 2020 (Figure 3a). In contrast,
municipal surface water use in the middle region was on average 11% lower during the drought and 15% lower following the

drought (Figure 4b). Municipal surface water use in the upper region, while small in magnitude (Figure 3g), showed even
larger declines than the middle region (Figure 4b). A consistent pattern in municipal groundwater use shared by all three
regions was increased use during the drought followed by reduced use after the drought, suggesting temporary shift towards
groundwater to compensate for reduced surface water supply (Figure 2). Only in the lower region has municipal groundwater
use in the post-drought period remained higher than the pre-drought period, likely related to accommodating the large

population increase from 2008-2020.

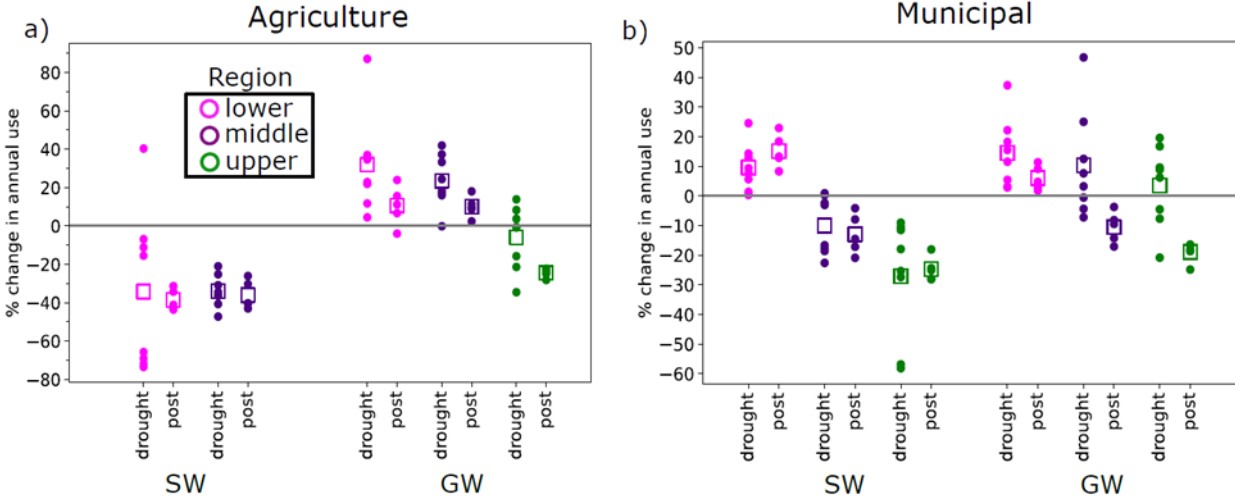

**Figure 4:** Change in agricultural (a) and municipal (b) surface water (SW) and groundwater (GW) use during the drought
(2008-2015) and post drought (2016-2020) periods compared to the pre-drought 2000-2007 period. Annual values are open
circles and time period means are open squares. No SW for agricultural use in the upper region is why it is omitted from (a).




### 3.2 Reservoir Drought

Due to the reliance on reservoir storage for water supply in the middle and lower regions, reservoir drought is a key aspect of
the drought and a nexus of multisector interactions (Figure 2). The magnitude of sectoral disruption and the speed that
reservoir drought develops depends on region-specific sectoral water demands and overall reliance on surface water. Prior to
the drought, the lower region used 5-6 times more surface water than the middle region (Table 1). At the onset of the
meteorological drought, middle region reservoirs were less than 50% full, and can be considered to already have been in the
midst of a long-term reservoir drought (Figure 1d), while in contrast, lower region reservoirs were completely filled (Figure
1e). Conditions only worsened in the middle region as the drought progressed and storage did not recover to the 2008-pre-
drought conditions until 2018. However, because of much lower agricultural surface water use (less than 1/10th of the lower
region), the middle region is not prone to large interannual variability in storage from supplying irrigators (Figure 1d).
Additionally, surface water use from other sectors (ex. municipal, thermoelectric) in the middle region is much smaller than
the lower region as well (Figure 3 e, h, k). In fact, the total surface water use in the middle region during 2000-2007 was
47% less than the municipal use alone in the lower region. In contrast to the gradual storage declines in the middle region
(Figure 1d), in both 2008-2009 and 2011, lower region reservoir storage declined more sharply, by over 40% during each
period (Figure 1e). Reservoir releases for surface water irrigation were the largest driver of large annual declines, but
significant municipal demand also contributed to storage declines during the most severe meteorological drought years.

The middle and lower regions both experienced sustained record low storage during the second half of the drought (2012-
2015). During this period, storage levels in the lower region fluctuated between 40-50% capacity and middle region between
10-20%. Based on the annual surface water use under drought conservation measures during the 2012-2015 period, storage
levels represented around two years of supply in each region. Low reservoir storage was the primary cause of agricultural
water shortages for surface water-dependent irrigators in the lower region, municipal water use restrictions in the middle and
lower regions, reduced hydropower generation, and exacerbated environmental flow and water quality issues (Figure 2). A
series of large precipitation events in 2015 ended the drought and replenished lower region reservoirs, which by 2016 were
completely full, while the middle basin storage only recovered to 25% capacity (Figure 1d). However, the middle region
considers 2015 the end of the drought of record, reinforcing the region-specific nature of reservoir drought impacts.

A specific feature of the 2011 to 2015 period that maintained severe reservoir drought in the lower region was the absence of
any large storm events to replenish storage. In 2011, inflows to lower region reservoirs were the lowest on record, and only
10.6% of average annual inflows during 1942 to 2017 (Austin Water, 2018). To contextualize how unprecedented 2011
inflows were, the lowest inflows during the 1950's drought were approximately four times greater than in 2011 (Austin
Water, 2018). Inflows to the lower region reservoirs continued at record-low levels from 2012 to 2014, all lower than the
worst year of the 1950's drought.





Evaporative losses further exacerbated reservoir drought (Figure 2). Mean annual evaporative losses in the basin are estimated to be 7.2% of reservoir capacity (Wurbs and Ayala, 2014). In 2011, lower region evaporative losses exceeded reservoir inflows, with an estimated 239 million m3 lost to evaporation — equivalent to ~10% of lower region storage

capacity and approximately the total annual municipal demand of the lower region (LCRA, 2022). Evaporative losses in the lower region ranged from 239 million m3 in 2011 to 135 million $m^3$ in 2014 (LCRA. 2022). Even at their lowest level in 2014, evaporative losses were equivalent to around two-thirds (60-70%) of lower region municipal surface water use.

Of the many factors that produced reservoir drought (Figure 2), the two most significant were 1) persistent record low

inflows and 2) large releases to agriculture in 2008-2009 and 2011. The decision to release large amounts of water to irrigators that accelerated the development of reservoir drought was based on decades of experience where storage typically recovered within a year or two of large storage declines. A permanent outcome of the drought was the adoption of more conservative reservoir management policies (Figure 2), discussed in Section 4.3.

**3.3 Agricultural Impacts**

Reduced agricultural production was one of the most disruptive impacts of the drought. Due to the direct dependence of vegetation health on soil moisture (Figure 2), agriculture is typically one of the earliest and most impacted sectors from meteorological drought (Van Loon et al., 2015). To illustrate agricultural impacts in the basin, county-level crop acreage and

production data from United States Department of Agriculture (USDA) was aggregated for four major crops (Figure 5). Cotton is a major crop in the middle and upper regions, winter wheat is mainly grown in the middle region, and corn and rice are only major crops in the lower region (Figure 5a).

The simultaneous stressors of increased plant water demand and physiological stress from high temperatures are the main

drivers leading to diminished yields and high abandonment rates in the region during the hot, dry drought conditions in 2008-2009 and 2011 (Figure 2) (Anderson et al., 2011; TWDB, 2022b; Nielson-Gammon, 2012). For all of the crops but rice, these years were generally associated with the lowest harvested acreage, production, and yield, (Figure 5 b-m) resulting in large agricultural economic losses (Figure 2) (Anderson et al., 2011; TWDB, 2022b). Compared to average agricultural gross domestic product (GDP) during 2000-2007, average GDP in the basin over 2008-2015 was $574 million lower (35%)

and in 2011 $913 million lower (56%) (all values inflation adjusted to 2022). The upper region was more severely impacted and disproportionally so due to its large agricultural sector. During 2008-2015 upper region agricultural GDP was reduced by 51%, while the middle and lower regions were only reduced by 26% and 24%. Agriculture comprises around 15% of the upper region GPD compared to less than 0.5% in the other two regions.

**Figure 5:** Locations of major crop production (a). Harvested acres (b-e), units produced (f-i), and yield (j-m) for the four crops. Crop-specific units of production: 480-pound bales for cotton, bushels for corn and wheat, and 100-pound units for rice. Cattle herd data for each region (n-p). This data includes all counties in each region.

The severity of impacts varied by region due to the spatial heterogeneity of drought (SI Figure 3) and differences in the proportion of irrigated versus dryland crops. Dryland farming is reliant on precipitation to meet plant water demand, and therefore is more vulnerable to meteorological drought than irrigated farms that can supplement precipitation deficits. Differences in the proportion of dryland cotton between middle (29% production irrigated) and upper regions (55% production irrigated) explain typically lower yields in the middle basin (Figure 5j), and larger reductions in production and harvested acres during the most severe drought years (Figure 4 b, f). Compared to 2010, in 2011 cotton acreage in the upper region declined by 64% while area in the middle region decreased by 87.5%. Cotton acreage and production gradually





recovered to pre-drought levels over 2012-2015. Winter wheat is another example of severe yield, acreage, and production declines for dryland crops (Figure 5 d, h, l). Prior to the drought, less than 10% of annual production was for irrigated wheat and even during the drought only 16% of production was irrigated. In 2009 and 2011, wheat production declined by 64% and 86% compared to the preceding year. Corn is also primarily dryland and had reduced production and yield in 2009 and 2011
but by 2013 production recovered to levels greater than before the drought (Figure 5 g, k). Corn continued to increase following the drought with post-drought area and production almost doubled relative to pre-drought levels (Figure 5 c, g).

Cotton is by far the largest and most significant crop in the basin. Cotton acreage is typically more than double the combined areas of winter wheat, corn, and rice (Figure 5). Texas is one of the major global producers of cotton and comprises a large
enough fraction of supply that the severely reduced production in 2011 contributed to the unprecedented price spike in cotton, which increased 153% between March 2010 and March 2011 (U.S. Bureau of Labor Statistics, 2011).

Rice differs from the three other crops because it is primarily irrigated by surface water flood irrigation. The decrease in rice production from 2012-2015 was a result of curtailment of lower region reservoir releases. 2012 was the first time in the
basin's history that agricultural water deliveries in the lower basin were curtailed, and curtailments continued from 2013-2015. Most of the of surface water for rice is classified as interruptible supply, which can be cut off if reservoir storage falls below trigger levels. Thus, the 60-70% reduction in rice production from 2012-2015 was a cascading impact of reservoir drought (Figure 2).

A potential adaptive response during drought is to temporarily switch to lower water demand, more drought-tolerant crops (Fisher et al., 2015; Glotter and Elliott, 2016). Temporary increase in sorghum production in the upper region is a potential example of crop switching (SI Figure 4). Increased sorghum, combined with decreased wheat and cotton also occurred during the 1950's Texas drought (TWDB, 2022b). Sorghum has lower water requirements and is more drought-tolerant than cotton or wheat (TWDB, 2022b). The largest single-year increase in sorghum production occurred in the upper region in
2008, while cotton production dropped by 55% compared to 2007, sorghum production increased by 350%. Sorghum production in the lower and middle regions did not show evidence of crop switching, and both regions display a long-term decline in sorghum production from 2000-2020 (SI Figure 4).

The drought also caused large reductions in cattle in the middle and lower regions, with a 17% (224,000) decrease from 2011
to 2012. Exceptionally low spring precipitation in 2011 prevented development of dryland crops for cattle feed and adequate forage growth for pasture (Figure 2) (Nielson-Gammon, 2012). Economic losses for ranchers were related to increased need to purchase feed, higher feed costs because of reduced availability, and lower sale price for cattle because the market was flooded by supply, as ranchers couldn't afford to maintain herd sizes (Countryman et al., 2016). Feed prices continued to increase in 2012 and 2013 due to the 2012 drought that impacted much of the Central U.S. feed supply chain (Countryman et





al., 2016). This reduced profitability for livestock caused ranchers to further reduce herd sizes (Figure 5 n-p) (Countryman et al., 2016). Cattle did not increase until 2015 and through 2020 herd sizes had not yet recovered to pre-drought numbers (Figure 5).

## 3.4 Environmental Impacts

### 335 3.4.1 Wildfire and Landcover

Drought increases wildfire risk by reducing plant moisture which increases the flammability of vegetation and likelihood of ignition, and increased flammability of parched vegetation can lead to more rapid spread and more intense burns (Figure (Littell et al., 2016). The dry and abnormally hot conditions in 2008 and 2011 (Neilson-Gammon and McRoberts, 2009; Neilson-Gammon 2012) were the two most severe wildfire years in the state during drought (SI figure 5), and the record dry

and hot conditions in 2011 produced the worst wildfire year in the state's history (Texas A&M Forest Service, 2011). 2011 accounted for 52% of the total area burned in the Colorado Basin over the drought period. However, the fraction of burned area in 2011 varied over the different regions with over 88% in the upper region, 50% in the middle, and 40% in the lower, and the two worst drought years (2008 and 2011) account for 57% in lower region, 88% in middle, and 90% in upper (SI Figure 5). The upper and middle regions are more arid and grasslands/shrublands, which were more impacted by hot/dry

drought-driven wildfires (Nielson-Gammon, 2012).

The record wildfires in 2011 are considered to have been partially a result of increased fuel, due to the wet year in 2010 that led to grass and shrub growth (Nielson-Gammon, 2012). A similar correlation has been observed in other Western US states where wet years followed by severe drought are often associated with increased wildfires (Scasta et al., 2016). The record

wildfires of 2011 are considered to be combination of 1) additional fuel combined with 2) increased flammability from extreme drought, and 3) unusually windy spring weather that enhanced wildfire spread (Neilson-Gammon, 2012). Firefighting cost for Texas were estimated at $48 million (Neilson-Gammon, 2012). Of the estimated $500 million in fire-related losses in 2011, $325 million (65%) was associated with the Bastrop Complex fire located in lower region city of Bastrop and remains the costliest fire in state history (Texas Standard, 2021).


In addition to vegetation loss from fires, the extreme dry and hot conditions during 2011 caused widespread tree mortality in the middle and lower regions, due to depleted deep soil moisture that typically buffers trees from short-term drought (Nielson-Gammon, 2012). Estimates indicate that there was an 8-10% canopy loss in the middle and lower regions (Schwantes et al. 2017). A statewide study by Moore et al. 2016 found single-year mortality percentages of 6-6.6% in the

middle region and 7.4-9.7% in the lower region, similar to the estimates from Schwantes et al., 2017. Crouchet et al. 2019 studied tree mortality in the middle region found a 9x increase in mortality compared to a typical year. The upper region was not affected by tree mortality because it is scrubland largely devoid of tree cover. Tree mortality also affected cities, with mortality rates in parts of Austin reaching 20% in 2011 (NASA, 2019). While the record hot, dry conditions in 2011 have



been the focus of most studies, Klockow et al. 2018 found pest-driven mortality increased during 2012-2015 in Eastern
Texas and hypothesized that this was related to physiological stress induced by 2011 combined with the continuation of
drought conditions.

**3.4.2 Streamflow, Surface Water Quality, and Environmental Flows**

Reduced streamflows (hydrological drought) caused primarily by prolonged and severe meteorological drought, were further
exacerbated by sectoral surface water use and reservoir management (Figure 2). To contextualize the severity of the
hydrological drought, streamflow at six locations in the basin are summarized using flow-duration plots (Figure 6 a-f).
Locations a-c are located along the mainstem of the Colorado River, while locations d-f are tributaries (Figure 6j). Figure 6
a-c additionally show the flow duration curves for the 2000-2007, 2008-2015, and 2016-2020 periods. The curves for the
pre-drought (2000-2007) and drought (2008-2015) periods were used to calculate percent reduction in flow over the entire
range of exceedance probabilities (Figure 6 a-f). Median to low flows are critical for stream habitat and water quality
(Caldwell at al., 2018; Konrad et al., 2008; Wineland et al., 2021) while high flows are important for replenishing reservoir
storage (Figure 2).

During the drought, flows along the mainstem were generally 40-60% lower across the spectrum of flow percentiles (i.e., the
high, median, and low flows were all heavily reduced), while the tributary locations had more heterogeneity in the nature of
their flow reductions. The San Saba location (Figure 6d) showed greater than 45% reduction across all flow percentiles,
while the spring-fed South Concho (Figure 6e) and Barton (Figure 6f) locations had less severe impacts to low flows (often
considered to be defined by the 90th or 95th flow exceedance percentiles). Prolonged hydrological drought can affect
groundwater levels, which can in turn affect streamflow by reducing groundwater baseflow and spring discharge (Smith,
2013; Smith et al 2015), demonstrated by reduced flows at e and f (Figure 6j).

Reduced streamflow caused surface water quality impacts in streams and lakes in the middle and lower regions, and even the
coastal estuary at the basin outlet (Figure 2). Water quality impacts included increased salinity, algae, metals, and nutrients
(nitrogen and phosphorus), which are surface water quality impacts commonly associated with drought (Mosley, 2015). One
way that reduced flows affect water quality is by increasing the concentration of pollutants in surface water. observed both in
point source pollution (ex. treated wastewater outflows) and non-point source pollutants (ex. runoff from agricultural or
urban land) (Mosley, 2015). The example we provide is for a segment of the Colorado River downstream of one of Austin's

**Figure 6:** Flow duration curves for the pre-drought (2000-2007), drought (2008-2015), and post drought (2016-2020) periods for three locations along the Colorado River, TX (a-c). Percent reduction in exceedance probability flow for the drought period compared to the pre-drought period (a-f) for six locations (3 for Colorado River and 3 for tributaries). Specific conductance data at two middle region reservoirs O.H. Ivie (Ivie) and Spence (Spen) (g) and two lower region reservoirs Buchanan (Buc) and Travis (Tra) (h). Nitrate and phosphorus data for the Colorado River downstream of Austin (i). Locations of discharge and water quality data (j) and denote symbols for subplots g and h that show data for two reservoirs.

two water treatment plants (Figure 6i), showing consistently elevated nitrogen and phosphorous concentrations during 2012-2015. Low streamflow also affected water quality in the Matagorda Bay estuary where the Colorado River discharges into the Gulf of Mexico. Discharge from the Lower Colorado River to Matagorda Bay in 2011 was 274 million $m^3$, representing a decrease of over 78% compared to the average annual discharge of over 1.2 billion $m^3$ between 1980 - 2010, marking the lowest on record since 1977 (TWDB, 2015). This historically low freshwater input resulted in increased salinity levels in the



estuary that reduced habitat suitability for oyster, crab, shrimp, and fish 1977 and impacted commercial fishing operations (TWDB, 2015).

Reservoirs and reservoir operations were related to a variety of surface water quality impacts (Figure 2). Due to the reservoirs being at critical levels between 2012 and 2015, environmental flow releases were reduced by about 86%, decreasing from 38 to 40.7 million m$^3$ in 2011-2013 to only 5.7 million m$^3$ in 2014, and there were no releases in 2015 (LCRA, 2022), almost certainly impacting downstream water quality and habitat. In the lower region, the drought led to elevated nitrogen levels in reservoirs that caused increases in microalgae population and a shift towards more harmful algae

strains (Gamez, et al. 2019), specifically cyanobacteria, which can produce harmful algal blooms (Beversdorf et al., 2013). Water quality in middle region reservoirs was impacted from naturally high levels of chlorides, sulfates, trace contaminants (ex. arsenic), and total dissolved solutes in groundwater baseflows (Region F, 2015). Due to hydrological drought, groundwater baseflow comprised a larger fraction of river flow, resulting in degraded water quality in the middle region. If groundwater has high solute concentrations or trace contaminants, the increased baseflow fraction during drought has been

shown to degrade surface water quality (Jones and van Vliet, 2018). Reservoir water quality was further degraded by evaporation that concentrates solutes. Specific conductance data (proxy for solute levels) for two key middle region supply reservoirs (O.H. Ivie and Spence) show concentrations steadily increasing from 2008-2013 (Figure 6g). Fresh inflows in 2013 substantially reduced solute concentrations in these reservoirs, though total storage in the middle basin had little change (Figure 1b). The two main lower region reservoirs (Buchanan and Travis) also show increasing solute concentrations during

the drought (Figure 6h), but their magnitude was much smaller and not a concern for potable water quality.

### 3.5 Economic Impacts

It is difficult to precisely quantify and directly attribute economic impacts to drought (Naumann et al., 2021; Stahl et al.,

2016). However, sectoral data on employment, GPD, and population growth at regional and basin scales enables a first-order assessment of whether any changes coincide with the drought period.

Population growth in the basin, including the rapidly growing Austin metro area, remained constant throughout the 2008-2015 period and did not show a reduced growth rate at any point during the drought (Figure 3 a-c), even during (2011-2015)

when strict conservation measures were in place. Additionally, key economic metrics of total GDP (Figure 7) and employment (SI Figure 6) both showed steady and sizeable growth throughout the drought. As shown in Figure 7, GPD decline in the middle and upper basins can be attributed to the oil and gas sector that is unrelated to the drought. While the drought had significant negative impacts on the agricultural sector GDP, agriculture represents a small fraction of total GPD and regional employment. Even in the upper basin, where 99% of water use is for irrigation, agriculture accounts for less





than 15% of jobs and 15% of GDP. However, agricultural impacts would have been far more severe if losses weren't partially offset by federal assistance and crop insurance (TWDB, 2022b).

**Figure 7:** Regional annual GDP for all sectors (a-c), agriculture (e-g), oil, gas, and mining (OGM) (h-j)., real estate (k-m), and all sectors minus oil, gas, and mining (OGM) (n-p).

Aside from agriculture, a specific sector harmed by the drought was the real estate market for lakeside homes, whose values are strongly tied to the recreational and aesthetic value of lakes. An analysis by Morris (2019) of home values around the lower region reservoir Lake Travis showed that the drought had large adverse effects on property values. Accounting for both loss of value and lost appreciation, lakeside homes incurred over $2 billion in estimated losses between 2011 and 2015



(Morris, 2019), whereas the real estate market in Austin and lower region exhibited strong growth throughout the drought
(Morris, 2019) (Figure 7).

## 3.6 Energy Production

The power sector notably did not suffer any major adverse impacts during the drought (TWDB, 2022b), and there were no
reports of significant outages even during record drought in 2011 (Scanlon et al 2013a). The absence of substantial reliance
on hydropower in the basin (less than 3% of annual production) resulted in no *significant* impact on power generation from
curtailed reservoir releases (Figure 8).  Additionally, many thermoelectric plants in the basin had already transitioned to low
water demand cooling technologies prior to the drought and thus were "pre-adapted" for severe and prolonged drought
conditions (Scanlon et al. 2013a). Natural gas facilities with high water efficiency technologies such as combustion turbine
and combined cycle (with cooling tower) are prevalent in the middle and upper regions (Scanlon et al 2013b). There is only
one high water demand coal plant in the lower basin, which is supported by a guaranteed firm water contract from lower
basin reservoirs (LCRA, 2022). Many of the thermoelectric plants also have their own reservoirs, including the South Texas
Nuclear Plant in the lower region, which provide more reliable supply than solely relying on run-of-river diversions. These
factors highlight the significance of institutional arrangements and engineered water infrastructure for reducing power sector
vulnerability to drought.

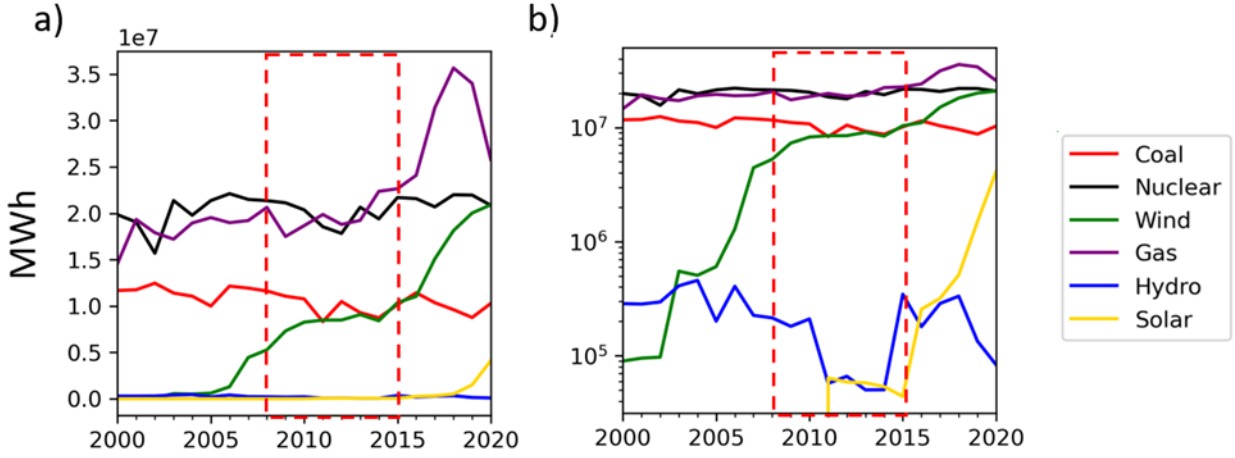

**Figure 8**: Electricity generation by fuel type for power plants in the Colorado Basin, TX. Annual generation in linear (a) and
log10 (b). Data from Energy Information Administration (EIA, 2022).

Over the course of the drought wind power production in the basin almost doubled (98% increase), mostly in the water-
scarce middle and upper regions, and by 2015 was similar in magnitude to coal power production in the basin (~10 million





megawatt hours (MWh) (Figure 8). Solar power did not have large growth until after 2015, but between 2015 and 2020 production increased from 44,000 MWh to 4.1 million MWh (Figure 8). Combined wind and solar production by 2020 (2.5 million MWh) was more than double coal power and on par with gas power production in the basin. An advantage of wind
and solar power in a water-stressed region is electricity generation with zero water requirements. This is an example of how decarbonization and energy transitions can reduce water reliance and water-supply vulnerability of the power sector (Byers et al., 2014, Zohrabian and Sanders, 2018). However, new vulnerabilities can emerge with increased reliance on renewables, such as periods of reduced wind speeds if a large fraction of regional supply is sourced from wind power (Wessel et al. 2021).


**4. Adaptive Responses to Extreme Drought: Insights from Water Planning and Management**

Drought often drives management responses and innovation (Lund, et al. 2018; Van Loon et al., 2016). To understand the substantive ways that the drought shaped water supply planning, we conducted a comprehensive review and analysis of data
in regional water management plans from 2011, 2016, and 2021 for each of the three regions in the basin (Region F, 2010, 2015, 2020; Region K, 2010, 2015, 2020; Region O, 2010, 2015, 2020). Our analysis was additionally supported by publicly available reports from utilities and municipalities in the basin.

Regional water plans are issued on a 5-year planning cycle and have been mandated by state law since 1997 in response to
severe drought conditions in 1995 and 1996 (Wurbs, 2015). An advantage of the relatively short 5-year planning cycle is the ability to respond to recent changes in water availability and sectoral demand. However, Nielson-Gammon et al., 2020 point out that a current blind spot of the regional water planning methodology is the "rear-view" drought of record approach that uses the worst historical drought as the basis for determining future water needs. Using the "drought of record" framework, water supply needs are based on shortages that would occur under a repeated drought of record event. Future shortages are
calculated based on the difference between projected future demands (based on estimated sectoral growth) and available supply under drought of record conditions.

The 2011 plans were developed during 2007-2010 before the most severe impacts had occurred and prolonged drought had set in, the 2016 plans were developed after the basin had experienced record drought in 2011 and unabating drought
conditions from 2012 to mid-2015, and the 2021 plans were created with full understanding of the new drought record. The drought resulted in large increases in proposed investments to meet long-term water needs, with the largest increase in planned projects in the lower region ($3.63 billion increase from 2011 to 2016 and an additional $623 million from 2016-2021) and moderate increases to the middle basin ($281 million from 2011 to 2016 and an additional $410 million from 2016-2021) (regional costs converted to 2022 dollars). Notably, the drought did not cause any major changes in the upper




region due to its low sectoral demand outside of agriculture and there is no economically viable alternative irrigation supply
other than continued use of groundwater.

**4.1 Drought of Record Impact on Future Sectoral Water Supply Planning**

Our review tabulated recommended additional water supply for sectors in each region along with the estimated sectoral
shortage in a repeated drought of record (Figure 9). Most of the future supply needs and recommended additional supply are
associated with the municipal and agricultural sectors (Figure 9), the two largest sectors in the basin.

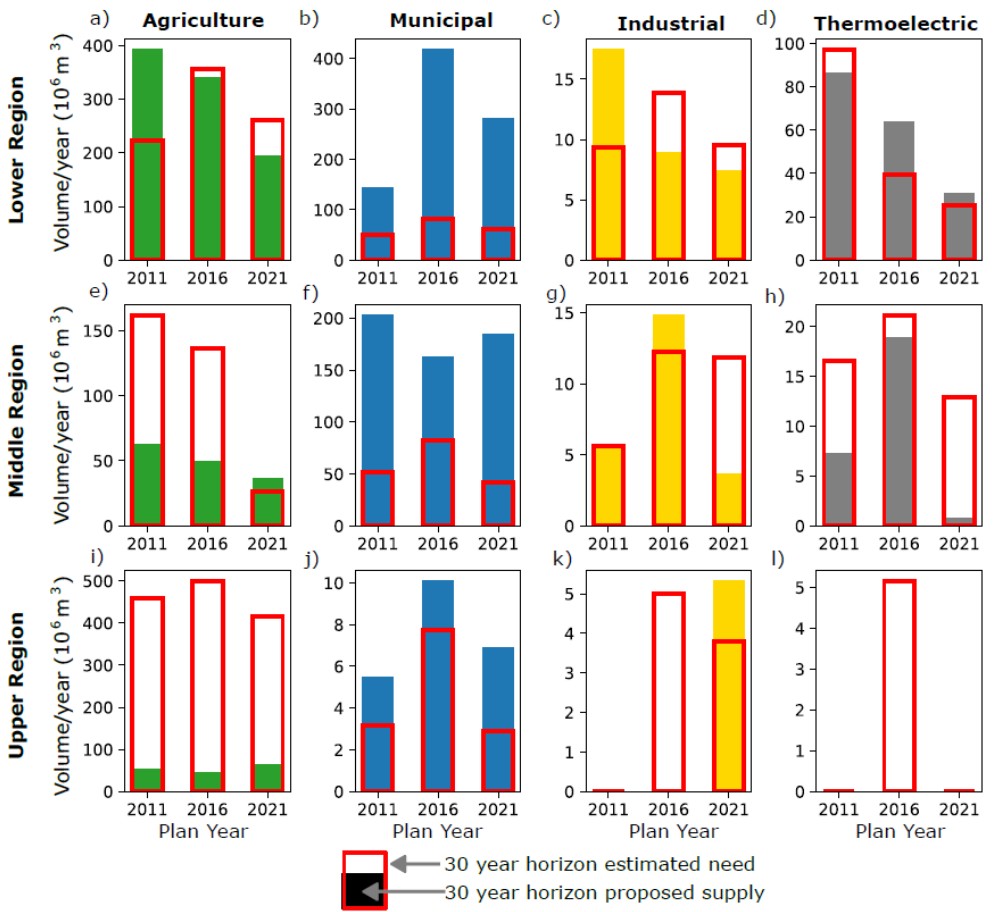

**Figure 9**: Filled bars show 30-year additional recommended supply (acre-feet/year) for each sector within each region, while
unfilled red bars are estimated annual sectoral needs under a repeat of the drought of record in the same 30-year horizon.

A nearly 300% increase in planned municipal supply volume for the lower region between 2011 and 2016 was the largest
planning response in the basin (Figure 9b). A consistent pattern across all regions is that recommended municipal supplies





far exceed projected future needs, which is intended to serve as a sizable buffer or 'safety factor' should a future drought be
more severe than the historical reference, considering the use of the drought of record methodology. In contrast,
recommended agricultural supplies typically do not exceed projected needs, and in the case of the upper basin are a small
fraction of future needs, and reflect anticipated reduced long-term supply from groundwater depletion (Region O, 2020) and
the lower priority for accommodating agricultural shortages. Proposed additional supply for thermal electric power meets
lower basin needs, but not middle and upper basin needs. However, the middle basin noted that the power plants included in
the regional water plans are being phased out in the near-future and that projected 30-year demand is not accurate; the upper
basin need in 2016 appears anomalous.

## 4.2 Water Management Strategies to Meet Future Supply Needs

Our analysis also compiled the specific sources of additional supply to meet the recommended supply targets for each
planning region (Table 3). We identified 13 water supply strategies proposed to meet future water needs in the basin (Table
3). The strategies can be classified into one of the following three groups: 1) demand reduction, 2) creation of new supplies,
and 3) alternate use of existing supplies. The three regions have notable differences in what combination of the 13 strategies
they propose using to meet projected needs under a repeated drought of record, which reflect different sectoral needs,
available supply sources, and strategy cost.

| | | Demand Reduction | | Existing Supplies | | New Supplies | | | | | | | | |
| Year | Region | Conservation | Drought Management | Voluntary Transfer | Subordination | ASR | Brush control | Desal | Groundwater | New Reservoir | Return Flows | Reuse | Rain Harvesting | Advanced Treatment |
|---|---|---|---|---|---|---|---|---|---|---|---|---|---|---|
| 2011 | Lower | 219.7 | 0.0 | 0.0 | 0.0 | 0.0 | 0.0 | 8.1 | 108.4 | 0.0 | 35.3 | 72.5 | 0.0 | 0.0 |
| 2016 | Lower | 256.4 | 182.1 | 0.0 | 0.0 | 64.3 | 4.2 | 0.0 | 32.5 | 151.5 | 54.5 | 72.2 | 10.2 | 0.0 |
| 2021 | Lower | 194.0 | 93.1 | 0.0 | 0.0 | 20.5 | 2.6 | 0.6 | 35.6 | 34.6 | 52.5 | 64.2 | 3.9 | 0.0 |
| 2011 | Middle | 67.8 | 0.0 | 25.7 | 93.5 | 0.0 | 10.6 | 19.8 | 41.9 | 0.0 | 0.0 | 15.4 | 0.0 | 0.0 |
| 2016 | Middle | 66.4 | 0.0 | 21.1 | 63.7 | 6.2 | 27.7 | 8.8 | 20.7 | 0.0 | 0.0 | 15.9 | 0.0 | 15.4 |
| 2021 | Middle | 41.2 | 0.0 | 1.6 | 55.0 | 0.0 | 0.7 | 0.0 | 71.1 | 0.0 | 0.0 | 11.0 | 0.0 | 44.0 |
| 2011 | Upper | 56.4 | 0.0 | 0.0 | 0.0 | 0.0 | 0.0 | 0.0 | 2.9 | 0.0 | 0.0 | 0.0 | 0.0 | 0.0 |
| 2016 | Upper | 49.0 | 0.0 | 0.0 | 0.0 | 0.0 | 0.0 | 0.6 | 5.1 | 0.0 | 0.0 | 0.0 | 0.0 | 0.0 |
| 2021 | Upper | 64.2 | 0.0 | 0.0 | 0.0 | 0.4 | 0.0 | 0.0 | 10.4 | 0.0 | 0.0 | 0.0 | 0.0 | 0.0 |

**Table 3**: Planned sources of additional supply ($10^6$ m$^3$) for planning regions in the Colorado Basin. ASR – Aquifer Storage
and Recovery; GW – groundwater.





### 4.2.1 New Sources of Supply Following the Drought of Record

The drought prompted planning regions to consider new sources of water supply. The 2016 regional water plans had six new supply strategies not present in 2011 plans: aquifer storage and recovery (ASR), rain harvesting, advanced water treatment, construction of new reservoirs, and brush control. Use of municipal return flows was not a new strategy in 2016, but the volume was increased by over 50% in the 2016 plan so is included in this section as well.

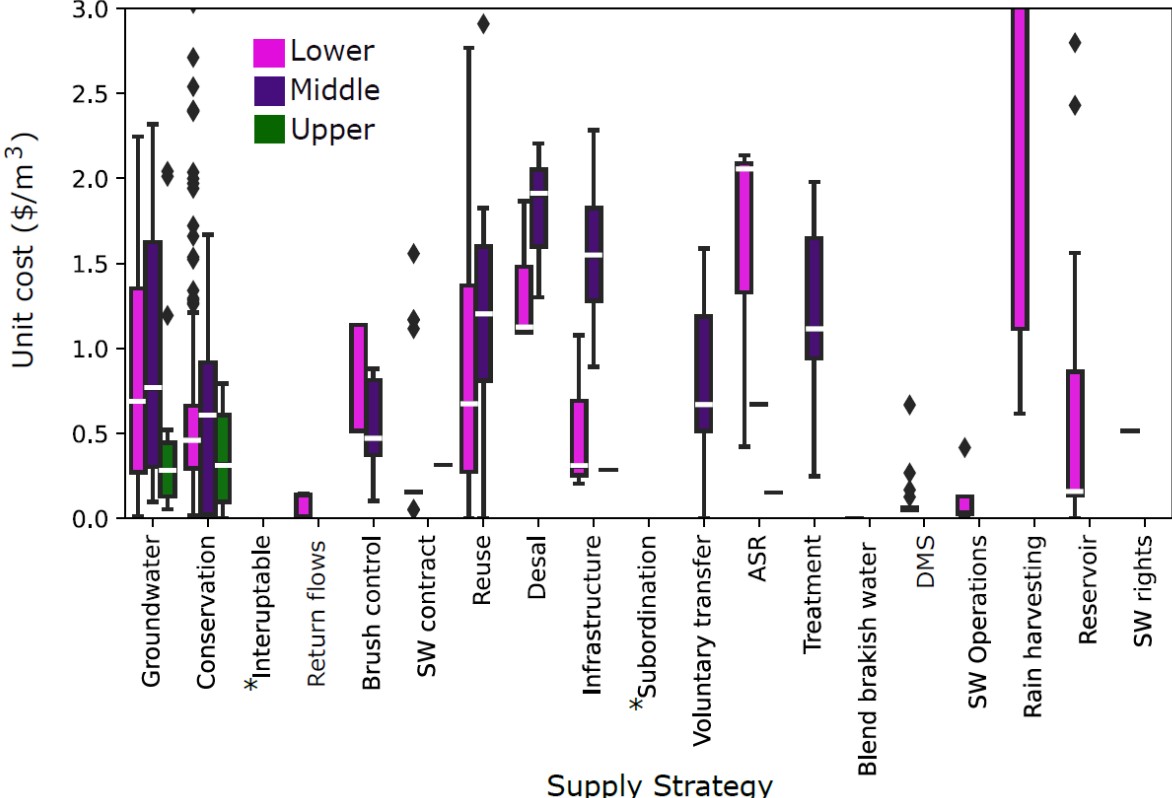

**Figure 10**: Unit cost per cubic meter for water supply strategies compiled from 2011, 2016, and 2021 regional water plans. Costs converted to 2022 dollars. ASR = Aquifer Storage and Recovery, DMS = Temporary Drought Management Strategies. No unit cost reported for interruptible supply or subordination. Boxes show interquartile range and median is shown by white line

These strategies have a wide range of unit cost (Figure 10, with return flows being the least expensive while advanced treatment, rain harvesting, and ASR generally the most expensive. ASR is primarily a strategy in the lower region, and likely due to its high estimated unit cost (Figure 10) was scaled back in the subsequent 2021 plan (Table 3). Advanced treatment is unique to the middle region and refers to upgrading existing water treatment facilities and building new facilities that can





treat surface and groundwater supplies to meet drinking water standards. Expanded advanced treatment capacity would
enable the middle region to use groundwater sources that currently exceed standards and treat reservoir water that can exceed
standards during periods of drought (Region F, 2015). The use of return flows in the lower basin is primarily for Colorado
River diversions downstream of Austin, but one project proposes to import municipal return flows from outside of the basin.
The drought accelerated the construction of an off-channel reservoir in the lower basin (Figure 2) that had previously been
an alternative recommended strategy in 2011 with a proposed implementation in 2030. The 111 million m$^3$ reservoir is
designed to make diversions from the Colorado River during high flow events to capture water that would otherwise flow to
the Gulf of Mexico. Planned to be fully operational by 2024, it is the first new major reservoir in the basin in decades and is
the most significant infrastructure project to increase supply of the lower basin. Brush control refers to the selective removal
of high-water demand plants (juniper, salt cedar, and mesquite) aimed at increasing groundwater recharge and reducing
riparian and shallow groundwater ET. Brush control was scaled back as a strategy in the 2021 plans and is not currently
proposed as major source of supply.

### 4.2.2 Supply strategies that Remained the Same or Decreased

Planned supply from groundwater and reuse remained the same or decrease after the drought. However, while groundwater
supply was reduced there was an increase in ASR, suggesting efforts towards more sustainable groundwater use. Reuse and
groundwater have a wide range of estimated costs (Figure 10). Reuse costs vary depending on whether the reuse is indirect
or direct and the intended end use, with potable reuse being more costly than non-potable reuse, in agreement with Cooley et
al. 2019. Currently active non-potable reuse in the basin provides supply to municipal irrigation (parks, golf courses), oil and
gas operations in the middle basin, and water for thermoelectric plants (middle and lower regions). The first direct reuse
facility in Texas became operational in the middle region city of Big Spring during in 2013. The Big Spring direct reuse
facility blends reclaimed water with raw reservoir water that is then treated in water treatment plant, providing 2.32 million
m$^3$/year of supply (Region F, 2015).

Estimates of new groundwater supply costs vary from 0.3 to 0.7 $/m$^3$ for the lower quartile to over 1 $/m$^3$ for the upper
quartile (Figure 10). Major cost factors are proximity to the groundwater source and end use. The top quartile costs are
associated with municipal supply projects developed far from the groundwater source that require extensive conveyance
infrastructure, whereas the lower costs are associated with local supplies associated with existing wellfields or non-municipal
use. An example of a high-cost, municipal supply groundwater project is the T-Bar Groundwater Well Field for City of
Midland (middle region) that became operational during the drought. The project added 13.8 million m3/year of supply at
cost of $209 million. Infrastructure included the installation of 43 wells and a 95 km 1.2 m diameter pipeline to convey
groundwater from the T-Bar Ranch, located outside the basin, to the city of Midland. Estimated unit costs for the project
were 1.15 $/m$^3$ (2008) per acre-foot during amortization (first 20 years) and 0.28 $/m$^3$ after (Region F, 2015).



Two strategies unique to the middle region are the use of existing supplies through voluntary transfers and subordination.
The recent drought of record reduced supply from these strategies by 50% (Table 3). Voluntary transfers are the temporary sale of surplus surface or groundwater supply between users within the middle region. Following the drought, available supply from voluntary transfers was reduced by over 90%. Subordination refers to junior water right holders in the middle region purchasing water from more senior downstream rights in the lower region. Under a strict priority system, junior middle basin water rights would not be allowed to make diversions during a drought of record due to legal priority of senior
downstream users. However, the middle and lower regions have historically cooperated to ensure adequate essential supply for junior middle basin users in critical sectors (ex. municipal and power) and anticipate continuing to do so in the future (Region F, 2020). However, estimated supply provided by subordination was reduced by 40% following the drought due to reduced estimates of firm (reliable) supply for the lower region.

**4.2.3 Conservation Strategies**

Demand reduction through conservation is a key strategy to meet future demand in all regions and was already a major strategy before the drought (Table 3). Conservation strategies were proposed across all sectors, with the largest conservation savings for municipal and agricultural sectors. Our analysis found that conservation is often more costly than many existing
supplies but is typically less expensive than developing new resources (Figure 10). Municipal conservation approaches include replacing water fixture efficiency, incentivizing low water landscaping, implementing permanent watering schedules (ex. Austin has year-round outdoor schedule, or limiting outdoor use during hot months May 1 to Sept 30th), improved metering, pipeline leak detection and repair, public outreach and education, customer engagement software (custom water use reports and water saving suggestions), and landscape standards for new development (Austin Water, 2018; Region K,
615 2020).

Austin has already implemented aggressive conservation measures, which have produced large, sustained reductions in per capita use (Figure 11). In 2010, Austin's water utility published a plan to reduce per capita use to 529 L/day by 2020 (Austin Water, 2010). The drought served as an accelerator of this objective. Per capita use fell to below 529 L/day in 2013, seven
years ahead of schedule, and the 76-113 L/day per capita reduction achieved during the drought has been sustained in the five years following the drought (2016-2020). Steep and lasting reductions in per capita use were achieved through an array of measures such as education, rebates for installation of drought tolerant landscapes, new ordinances for irrigation systems in new developments, rate increases, and rebates for water efficient fixtures (Austin Water, 2018).

Agricultural irrigation conservation measures include lining of canals, conversion of canals to pipelines, laser-levelling flood irrigation fields (primarily rice in the lower region), increased efficiency (conversion of flood to sprinkler and sprinkler to


drip), and real-time metering and monitoring (supports more accurate billing and data to support conservation improvements) (Region F, 2020; Region K, 2020; Region O, 2020). Colaizzi et al., 2009 specifically looked at irrigation conservation measures in the Ogallala (upper region) and found the most effective to be expanding use of weather-based
irrigation scheduling, converting flood irrigation to center pivot, and replacing high water demand crops like corn with lower demand crops like cotton.

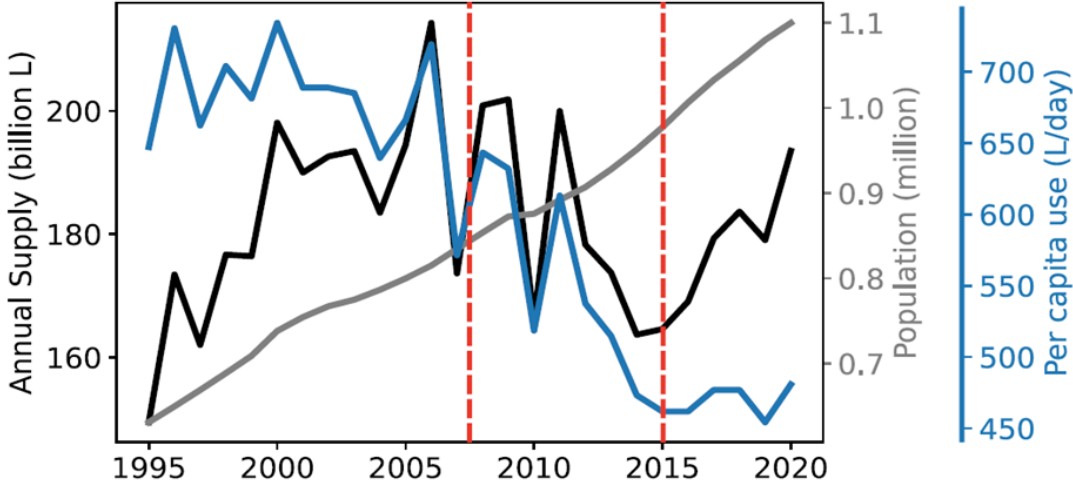

**Figure 11**: Austin Water annual water use (black), population (gray), and per capita water use (blue) from 1995 to 2020. Drought period shown by dashed red lines. Data Austin Water, 2022.

Temporary demand management is not unique to the lower region but is only explicitly accounted for as a source of supply to offset shortage during a repeated drought of record in the lower region. Most temporary demand management efforts are
aimed at reducing municipal outdoor use, which is a substantial fraction of total water demand, especially during summer months, and can be highly responsive to temporary reduction measures (Hogue and Pincetl, 2015). Temporary demand management measures include limitations on frequency, timing, and method of outdoor water use. These measures are only implemented under pre-defined drought trigger thresholds such as reservoir storage thresholds (ex., lower region storage below 60%) and peak daily municipal demand thresholds (ex., 120% of average daily demand) (Austin Water, 2016).
Outdoor water restrictions in the U.S. during drought have been shown to reduce residential water demand by ~20-50% (Gober and Quay, 2015; Mayer et al., 2015).





 **4.3 Water Management Responses and Planning Innovations**

Notable changes to water management and planning include updating policies to conserve water more aggressively during future droughts, new laws to improve regional water planning, and modelling advancements to improve water management and planning.


Following the drought, the lower region, highly reliant on reservoir storage, has implemented more stringent supply reduction triggers to conserve storage. Before the drought, available interruptible (non-guaranteed) supply was gradually reduced between reservoir storage thresholds of 70% to 15% capacity, and there were no restrictions to firm customers (Region K, 2010). Following the drought, operating rules were revised so that interruptible supplies can now be fully
curtailed below 45% capacity (Region K, 2020). Another major change is that lower region municipal firm customers now have drought trigger thresholds at 70% and 45% storage capacity that require corresponding use reductions of 5% and 10-20% (Region K, 2020). Under a scenario worse than the drought of record, firm customers will be subject to a minimum 20% reduction and are encouraged to use alternate supplies (ex. groundwater) (Region K, 2020).

There were also notable modelling capability improvements during and following the drought. The LCRA who manages lower region surface water supplies added new capabilities of their medium range forecast model used to inform reservoir operations. New features include revised reservoir operating rules, modification of environmental flow requirements, and the incorporation of El-Niño Southern Oscillation forecasts (Anderson and Walker, 2017). A Distributed Hydrology Soil-Vegetation Model (DHSVM, Wigmosta et al., 1994) model is under development for the basin that can produce high-
resolution naturalized flow inputs to either the official state Water Rights Analysis Package (WRAP) model (Wurbs, 2020) or the LCRA Riverware (Zagona et al., 2001) operational model for water management modelling studies. The DHSVM model will enable historically based drought of record analysis and also future climate scenarios driven by downscaled global climate model inputs. A modelling advancement implemented in the middle region was to represent sectoral water demand reductions during drought of record conditions (Region F, 2015). This modification to the WRAP model was aimed
at improving estimated water supply needs by better representing reductions in water demand during drought conditions.

The record drought also prompted Austin, the population hub of the basin, to more rigorously evaluate the long-term security of its water supply. In 2014, the Austin Water Resource Planning task force, created in 2007, recommended that the city perform its own independent assessment of water supply for the next 100 years (Austin Forward, 2018). The task force
recommended assessments occur on 5-year planning cycles, similar to the regional and state water planning cycles. The first long-term study for Austin was published in 2018 (Austin Forward, 2018). A notable feature of the study is the incorporation of future climate uncertainty into the assessment of Austin's long-term water supply, instead of the drought of record approach used in the state regional water planning.





Several state laws were passed, both during and following the drought, targeted at improving water planning. In response to numerous threats to municipal supplies in 2011, the 2012 state legislature passed TAC 357.42(d) requiring each regional planning group to collect information on existing emergency water connections. The law mandates each region to create and maintain a database of emergency supply connections and available supply volume of each connection. Before 2016, recommended water management strategies from previous regional plans were not tracked to determine their implementation

status. Starting in 2016, the TWDB requires each region to conduct a region-wide survey to track the implementation status of all water management strategies recommended in the previous plan. More recently, HB 807 passed in 2019, is designed to increase regional cooperation in water planning and promotes water supply from ASR by requiring all regional water plans assess ASR as a strategy (Kramer et al., 2019). While there are currently only six active ASR sites in the state, ASR is considered a promising long-term strategy for conserving groundwater resources. Notably, two of the six active ASR sites in

Texas are in the lower region of the Colorado Basin and multiple ASR projects were proposed in the 2016 and 2021 plans for the lower and middle regions (Table 3). HB 807 also requires the TWDB to create an Interregional Planning Council to improve coordination and share best practices between each planning region (Kramer et al., 2019).

## 5. Discussion

### 5.1 Long-Term Water Supply Challenges Facing the Basin

The combination of growing population, the possibility of more severe and prolonged droughts, and an anticipated shift towards hotter, more arid conditions pose significant long-term water management challenges (Banner et al., 2010). In addition to physical limitations on new supplies (ex. aquifer storage and capacity, stream flows, reservoir storage), laws and

regulations governing surface water and groundwater use also limit options for expanding water supply. Thus, the basin faces the challenge of finding additional reliable supplies when much of the easily accessible and low-cost surface and groundwater has already been appropriated and developed (Tidwell et al., 2014).

More arid conditions are anticipated to induce changes in soil moisture (Nielson-Gammon et al., 2020), potentially altering

runoff characteristics with important implications for water resources (Saft et al., 2015). A recent study found reductions in annual streamflow in the basin over the 2030-2100 period in almost half of global climate model scenarios considered (Austin Forward, 2018). A 20-30% reduction in water yield for the basin in the 21$^{st}$ century (runoff + groundwater recharge) was estimated in a CONUS-wide study by Brown et al., 2019. Observational data (1900-2017) already indicates significant downward trends in both streamflow and precipitation-streamflow ratios in the basin, with the strongest decline in the central

region of the basin (Harwell et al., 2020). Persistent record low surface inflows were a major contributor to the severe reservoir drought conditions from 2012-2015. Analysing the 2001-2009 Millennium Drought in Australia, Van Djik et al 2013 found that a median precipitation decline of 11% below average resulted in a 46% reduction in median streamflow



during the drought. This highlights the non-linear relationship between precipitation and runoff and the potential threat to surface water availability from even small reductions in annual precipitation.


A major challenge facing the middle and lower regions is that surface storage capacity is already maximized (i.e., there are no viable locations for additional major reservoirs), but population and associated surface-water reliant municipal demand are expected to continue to grow (Region F, 2020, Region K, 2020). A sobering statistic is that *lowest* per capita storage during the 1950's drought (previous drought of record) is approximately *equal* to the current maximum per capita storage

with every reservoir in the basin at full capacity (SI Figure 7). The buffer provided by reservoirs will be further diminished as the basin's population continues to grow.

Under current groundwater use conditions, only the upper basin is contending with highly unsustainable depletion (Scanlon et al., 2012, Region O, 2020). In the coming decades, agriculture in the upper region will have to adapt to reduced

availability from the Southern High Plains aquifer. While current middle and lower region agricultural groundwater use is also sizeable, groundwater availability models don't project that aquifers are being rapidly depleted like they are in the upper region (Region F, 2020, Region K, 2020).

## 5.2 Building a More Resilient and Sustainable Water Supply


Multiple recent studies have examined the 'reservoir effect' where regions with access to large reservoir storage can be prone to increased vulnerability to severe drought due to lack of supply diversification and lower incentivization for adaptive measures (Di Baldassarre et al 2018, Garcia at al., 2019). The recent drought exposed vulnerability of the lower region's reliance on surface water and reservoirs. Lund et al. 2018 explains that well prepared water systems typically avoid major

negative impacts, and that water management often improves after exposure to water scarcity. The planned diversification of water supply sources following the drought shown by our analysis (Table 3) indicate efforts to reduce reliance on reservoirs. Because of its chronically depleted reservoirs, the middle region was already adopting expanded groundwater, including out of basin groundwater imports, and unconventional supplies (direct and indirect reuse) earlier than the lower region.

Dependence on reservoir storage, and more generally surface water, can be reduced by increasing groundwater capacity and developing non-conventional water supply sources such as wastewater reuse, desalination (seawater and brackish GW), and ASR. Expanded groundwater capacity can offer a reliable supply for users confronted with more unpredictable surface water resources (Taylor et al., 2013). However, the location, scale, and frequency of groundwater use needs to be carefully evaluated, ideally to ensure that it is sustainable and that it will not adversely impact surface water baseflows (de Graaf et al.,

2019). Reuse has the benefit of creating additional supply close to the source of demand, low transmission costs, and low environmental impacts (Grant et al., 2012). However, increased reuse reduces water treatment plant return flows to



downstream users. This could be offset by more water being available to downstream users due to reduced upstream diversions, but the trade-off would need to be studied to assess the net impact. Potable reuse may have less environmental impacts and if often a cheaper unit cost compared to desalination (Hadjikakou et al., 2019). However, direct reuse faces 755 larger public perception challenges than indirect reuse or non-potable reuse (Lahnstener et al., 2018). ASR enables storage of surface water during periods of plentiful supply for later use and has the added benefit that stored water is not lost to evaporation. However, ASR is still a developing technology and has high abandonment rates due to a variety of issues such as well clogging, water quality, and insufficient recovery (ratio of injected to recovered supply) (Bloetscher et al., 2014). Managed aquifer recharge (MAR) has been employed since the 1960s and has seen significant growth in the last 30 years 760 (Dillon et al., 2019) is a lower-risk alternative to ASR to improve groundwater sustainability.

Equally important to expanding supply is reducing demand. Demand management encompasses a wide range of actions intended to reduce water use such as increasing efficiency, adopting or changing laws governing water use, and pricing strategies (rate-based), and is considered an essential component of water security (Cosgrove and Loucks, 2015). 765 Conservation is often much cheaper than development of new alternative supplies (Cooley et al., 2019), and was found to be a major component of agricultural and municipal supplies in the basin (Table 3). Demand management for agriculture includes government incentives for more efficient technologies (Fan et al., 2022, Region O, 2020), pumping fees per unit production, and total pumping limits (Hrozencik et al., 2017, Kumar et al., 2011, Rad et al., 2020). For municipal conservation, some research indicates that non-price approaches, such as restrictions, can be more effective than pricing 770 (Kenney et al., 2008), and the Dascher et al. 2014 analysis of consumer behavior in Texas during the 2008-2015 drought suggests that restrictions combined with outreach as most effective. Factors contributing to positive attitudes towards conservation include environmental awareness, education, and having experienced drought (Burton et al., 2007, Dickinson, 2001, Dascher et al., 2014). However, positive attitudes do not always produce behavioural changes (Gregory and Leo, 2003; Miller and Buys, 2008). Few people in the basin (citizens, water managers, politicians) experienced the devastating 775 drought of the 1950s, so the recent 2008-2015 drought was potentially a formative experience for the current generation of residents and demonstrated the value of conservation efforts for improving water security.

Our water supply cost analysis (Figure 10) showed that additional new supply tends to be more costly than existing conventional sources, particularly low-cost surface water. Historical development across the Western U.S. has relied on low-780 cost sources of unappropriated water or transfers of appropriated water (Tidwell et al., 2014). The increased cost of new supplies or conservation can be accommodated by and is justifiable for municipal and industrial uses, but costs of unconventional sources may be prohibitive for agriculture, where profit margins are slim (Hoppe, 2014). A common adaptive response to potential shortages in high-value sectors (municipal, industrial, energy) is to obtain supply from low value uses, typically from agriculture (Flörke et al., 2018). This practice raises questions about the magnitude of these 785 transfers on food security and regional agricultural production (Brown et al., 2019) and to what extent future water supply





will be offset by reductions to agricultural use. Improved management and conservation efforts in the upper region will only slow the timeline to depletion (Scanlon et al., 2012) and large declines in irrigated acreage are anticipated by 2100 (Deines et al., 2020).

**5.3 Impact of Drought to an Advanced Regional Economy**

The economic impact of drought relates to how dependent a region's economy is on water supply and access to trade to offset local impacts (Lund et al., 2018). Highly connected domestic and global trade networks in the 21st Century have greatly reduced the economic and societal impacts of drought (Lund, 2016, Lund et al., 2018). Water supply infrastructure
also buffers social impacts and economic disruption (Lund 2016). The combined factors of highly engineered regional water supply and domestic-global trade networks help explain why the drought did not hinder population and economic growth.

Our finding that the drought had little apparent overall effect on the basin-wide economy is in line with assessments of the 2001-2009 Millennium Drought in Australia (Van Djik et al., 2013) and the 2012-2016 drought in California, United States
(Lund et al., 2018). During the 2012-2016, California experienced a 1/3 reduction of water supply but only incurred economic loss equivalent to 0.09% of its economy (Lund et al., 2018), while the Millennium Drought in Australia reduced total GPD by only 0.4% (Van Djik et al., 2013). Recent examples from California, Australia, along with this study, demonstrate how decoupled modern economies are from the agricultural sector. Tubi (2020) terms this a shift from "climate sensitive" to "climate insensitive" economies. They analysed drought impacts in Israel from 1954 to 2017 found that Israel
transitioned from a climate-sensitive economy with large percentage GDP and employment in agriculture, to a climate insensitive economy over the 1960s and 1970s, where presently agriculture is less than 2% of GDP and employment (Tubi, 2020). However, it should be acknowledged that agricultural comprises a much larger fraction locally and regionally as exemplified by the upper region of the Colorado Basin, TX where it accounts for 15% of the economy and is also critical to food security and the broader rural economy.  However, our findings along with Van Djik et al., 2013 and Lund et al. 2018,
suggest that catastrophic drought would be required to substantially reduce the GDP of a modern economy.

**6. Conclusions**

Our analysis showed that the drought produced an array of environmental impacts, harmed agriculture, threatened water
supplies, and permanently altered water planning and management. Water supply infrastructure (reservoirs, pipelines, canals, and wells) and temporary demand management responses averted severe shortages to non-agricultural sectors. Evaluation of regional water management plans showed that the drought substantively affected water management planning with large increases in the variety of water supply strategies and municipal supply volume following the drought. Our review found that there is no "silver bullet" solution for the basin such as building a large new reservoir to accommodate future growth and



reduce vulnerability. Instead, a mosaic of supply and demand management strategies are needed to achieve long-term water security. Sustainable long-term water supply requires a combination of technological and management innovations (Gleick, 2018). Cosgrove and Loucks (2015) posit that the human component of water management (ex. political will, consumer willingness to pay) poses a larger obstacle to achieving sustainability than the technical aspects. Following the drought of record, there is evidence of proactive changes to water management and planning, with more sophisticated water supply

models being developed and used, more conservative drought management policies enacted, and several new laws regulating water planning. However, the difficult and key task of implementing the expensive water supply projects (over $6 billion in 2022 dollars) is largely yet to be accomplished. Water planning faces deep uncertainty about future demand (sectors, location, quantity) and availability of supply (quantity, reliability), and therefore it is imperative that both technical and institutional management approaches evolve as better data and modelling techniques become available. As indicated in the

title, we feel this study offers a "blueprint" that can be followed by other regional drought analyses. Our hope is that this work will inspire other comprehensive, multisectoral drought impact studies that improve our understanding of how regional nuances in climate, hydrology, ecosystems, institutional management, water supply infrastructure, and sectoral demand lead to specific drought impacts/risks and how these factors influence adaptive planning.

**Code availability**

The Python scripts for processing and plotting the data presented in the figures will be made available on an associated GitHub repository,

**Data availability**

All data presented was obtained from publicly available sources. All data presented in the text and supplemental figures will be made available on an associated GitHub repository.

**Author contribution**

SF, ST, NS, JR conceived the idea for the study. SF performed the data curation and formal analysis. SF and ST wrote the

manuscript draft. SF, NS, ST, BS, and JR reviewed and edited the manuscript.

**Competing interests**

The authors declare that they have no conflict of interest.

**Acknowledgements**

This research was supported by the U.S. Department of Energy, Office of Science, as part of research in MultiSector Dynamics, Earth and Environmental System Modeling Program. Pacific Northwest National Laboratory is a multi-program



national laboratory operated by Battelle for the U.S. Department of Energy under Contract DE-AC05-76RL01830. All of the data and code supporting this paper is available at: WILL ADD LINK UPON ACCEPTANCE.

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
