# Peer review of "Multisectoral analysis of drought impacts and management responses to the 2008-2015 record drought in the Colorado Basin, Texas"

_Natural Hazards and Earth System Sciences, 2023_

## Author Response (AR1)

**Reviewer 1:**

The manuscript addresses a very significant issue of regional multisectoral drought impact assessment. However, the manuscript is overly descriptive, resembling a dissertation thesis. Each section should be concise and engaging.

We thank Reviewer 1 for reviewing our manuscript and providing constructive feedback.

The feedback from yourself and Reviewer 2 that the organization and length of the paper hamper reader engagement prompted us to assess how the paper could be re-organized to communicate our findings more effectively. We implemented three major changes that we believe significantly improve the manuscript and address the concerns from both reviewers:

1. We expanded the Data section (formerly 2.3) to cover Data and Methods (now Section 2) that provides a formal description of the methods. Material related to the methods that was in former sections 3.0 and 4.0 have been moved to Section 2.
2. We moved the influence diagram (formerly Figure 2) to the *end* of the Results section. During our review of the paper in response to reviewer comments, we realized that introducing the full influence diagram at the beginning of the Results was likely overwhelming. The revised paper now introduces a highly simplified diagram in the Data and Methods section. The detailed influence diagram is now presented in a new Results (Section 3.3).
3. We completely rewrote the Discussion. Reviewer feedback helped us realize that the Discussion was too focused on introducing new information rather than highlighting and discussing key findings. We replace the previous three sections of the Discussion with two entirely new sections that focus on 1) insights into multisectoral dynamics and impacts, and 2) limitations and future work. The influence diagram is now used to guide the first Discussion section about key findings on how drought propagates different multisectoral impacts and their interactions.

Before providing point-by-point responses, we want to address the general comment that the manuscript is overly descriptive. We acknowledge in our responses below that some areas of the manuscript could indeed be more focused and succinct, and tightening the text was a key focus of our revisions. Trying to address the *why* of our findings led to the lengthy and descriptive style of writing that you point out. During revisions we made a concerted effort to shorten areas of the Results as long as doing so did not remove context of why the impacts or responses manifested the way that they did.

Some specific comments are as follows:

Figure 1: The legend needs correction, and the figure title is too long. Consider moving the region description to a separate paragraph.

Thank you for identifying the typo in the legend. We have fixed the legend in Figure 1 and shorten the figure caption by moving over half of the caption content to the main text of Section 1.1.

The Methodology and Results sections should be distinct for clarity. Currently, Section 3 combines both, making it challenging to discern the study's process and findings.

The revisions expanded the Data section (formerly Section 2.3) to include a formal description of the Methods by moving methods descriptions from Sections 3.0 and 4.0 to Section 2. The revised Results sections now exclusively focus on the findings, with all methodological details consolidated in Section 2.

In Section 3, there is an excessive explanation of what a DAG is and its benefits. However, it's unclear how you constructed the influence diagram, including the choice of nodes and links. Explain whether the influence diagram was adapted from previous literature or a survey. Elaborate on the process of selecting nodes and their relevance to the region.

In the interest of shortening the paper we have shortened the DAG explanation in Section 2 (Data and Methods). Regarding how the influence diagram was constructed, the revisions now explicitly state that the influence diagram is a *novel* product of this study (Lines 160-162). We created the influence diagram by synthesizing findings of drought impacts in the Colorado Basin, TX from reviewing thousands of pages of regional water planning documents, reading over a hundred academic papers and reports, and by analyzing the 15 datasets presented in this study.

In Figure 3, clarify the Y-axis for "g-i." Also, explain why there is a declining trend in agricultural water use for the upper region, if applicable.

We have renamed the Y-axis label for subplots g-i in Figure 3 to "Total sectoral use" instead of "Sectoral" which was vague. The declining trend that started during the drought does not have an obvious explanation because it does not reflect comparatively large reduction in irrigated acres for major crops. One plausible explanation would be adoption of more efficient irrigation technology, but we don't have data to support that hypothesis. We've clarified in Section 3.1 - the multisectoral water use that this trend is present but a reason for the trend is not supported by the crop data nor is explained in any of the sources we reviewed (Lines 199-203).

Elaborate on what you mean by "reflecting a transition to a more drought-tolerant supply" in line 185. Provide specific details or context to make this statement clearer.

We briefly expanded this statement, which is in reference to thermoelectric plants increasing their supply from GW (Lines 239-241).

The Results section is excessively lengthy, which results in poor readership engagement. Some of the figures may be moved as intermediary figures in the supplementary material, and essential findings should be emphasized in figures/tables in the manuscript.

Reducing the length of the Results section was something that we grappled with throughout multiple internal revisions before submitting to review. As mentioned in a previous response, the broad overview covering multisectoral drought impacts and adaptive responses covers vastly more topics than a typical research article. However, we don't want the length of the Results section to be off-putting to readers. Part of the revisions process was to shorten the Results

section as much as possible without jeopardizing the nuances of the findings. The content of the Results section has been reduced *by over 1,500 words* compared to the original version of the paper.

As part of the revision process, we evaluated the need to retain all 11 figures in the main text. We determined that the energy production figure (formerly Figure 7) was not necessary and have moved that to the Supplement (now SI Figure 6). The other figures are key to the Results and were retained.

The Discussion section reads more like a literature review than a discussion of the study's major findings.

Based on the feedback from yourself and Reviewer 2, the Discussion section was clearly a major weakness of the manuscript. One of the main focuses for our revisions was a complete overhaul of the Discussion. We replaced the three Discussion section with two new sections that focus on 1) insights into multisectoral dynamics and impacts and 2) limitations and future work.

The limitations and future scope of the study are not well depicted.

A discussion of limitations and future work has been added as a new section to the Discussion.

The reference format should be consistent.

We have carefully reviewed all of the citations to make sure they are correctly and consistently formatted.

**Reviewer 2:**

The paper "Multisectoral analysis of drought impacts and management responses to the 2008-2015 record drought in the Colorado Basin, Texas: A blueprint for regional multisectoral drought impact assessment" by Ferencz et al. present drought impacts in the Colorado Basin, Texas.

This topic is of great interest to the journal's audience and the research community. The authors assessed a wide variety of impacts that are detailed in the text. Despite the interesting topic, the paper presents some issues that need to be addressed before publication. The primary issue concerns the manuscript's style, which is very long and it is subdivided into paragraphs that tend to jeopardize the information provided.

We thank Reviewer 2 for reviewing our manuscript and providing constructive feedback.

The feedback from yourself and Reviewer 1 that the organization and length of the paper hamper reader engagement prompted us to step back and assess how the paper can be re-organized to communicate our findings more effectively. We have made three structural changes that we believe will significantly improve the manuscript and address the concerns from both reviewers:

1. Expand the Data section (Section 2.3) to also include a formal description of Methods by moving methods descriptions from Sections 3.0 and 4.0 to Section 2.

2. Move the detailed influence diagram (Figure 2) to the end of the Results section. We realized that introducing the influence diagram at the beginning of the Results can overwhelm the reader since it is a summary of all of the content that follows in the two Results sections. The revised paper instead introduced a highly simplified diagram that summarizes the content of each Results section and the detailed influence diagram was presented after section 4.0.
3. A complete overhaul of the Discussion. Your feedback helped us realize that the Discussion was mostly introducing new information rather than highlighting and discussing key findings. In the revision, we rewrote the Discussion sections to focus on 1) insights from the detailed influence diagram and 2) discuss limitations and future work, to guide the Discussion about how drought propagates different multisectoral impacts and their interactions.

The paper's division into paragraphs makes it fragmented and challenging to follow. While the information regarding drought impacts is highly relevant, its presentation is in the form of a list, which might make it difficult for readers to comprehend all the information. The paper would greatly benefit from consistent reorganization and synthesis of the text. Please consider using figures to visually synthesize the numerical data and trends, etc. For example, consider using figures to illustrate changes over time, as demonstrated in [1], which also published the dataset used for analysis.

Our interpretation of this comment is that it is referring to wanting a more condensed synthesis of findings. Our revisions aimed to synthesize the text in the Results sections, where possible, and were able to reduce the length of the Result by over 1,500 words. However, explaining impacts and responses across three different management regions inherently requires listing impacts/responses and explaining how they were similar or different between the three regions. As described in our response to Reviewer 1, over-synthesizing could remove the important context for the multisectoral impacts and responses. The results section is still longer than most papers but this is because we are addressing the content that would often be presented as separate papers – however, we think the reader greatly benefits from having the impacts and responses presented in a single coherent paper.

Regarding alternative data visualizations, part of the revision processes was to evaluate whether some of the current figures can be modified. We feel that the existing Figures are effective at displaying changes in key sectoral attributes over time (water use, crop production, GDP, surface flows and water quality). The figures in the example reference provided (Kriebich et al.) are effective at summarizing the *high-level global findings* from that work, but a critical aspect of this study is to present and discuss impacts and dynamics at the yearly time scale. A follow up study could present a more high-level synthesis for multiple basins in Texas that experienced the drought. Such a study would leverage the type of visualizations in Kriebich.

Lastly, the published paper provides a link to a GitHub repository that contains the datasets and data analysis and plotting scripts used to produce the figures in the paper.

There is missing information concerning the case study areas, particularly regarding the climate and the primary characteristics of the three areas. For instance, the Upper Region is characterized

by a low population, yet it utilizes significantly more water than the other regions. Notably, there is no storage associated with the Upper Region, and it primarily relies on groundwater. Could you please expand on these concepts?

We have expanded this section, keeping length issues in mind, to make sure that differences in regional characteristics are clear to the reader as they are key to understanding many of the drought impacts. Lines (61-89)

The Discussion section reports some data; please focus on the discussion alone, while the data should be presented in the main text.

As described above, there was a complete overhaul of the Discussion. The new Discussion sections do not introduce new data.

Please expand the conclusions and demonstrate the relevance of this work, not only specifically to the case study area but also more broadly in terms of lessons learned and potential future improvements for effective drought mitigation.

Reviewer 1 had a similar comment. The revised Discussion sections discuss key lessons learned from this regional study that can more broadly apply to improving drought mitigation in other regions.

Additionally, please provide figures in high-quality formats.

The figure quality from the Word to PDF conversion is very poor. We have provided high resolution SVG versions of each figure to the publisher and the high-quality versions will be used for the eventual publication.

Fig. 1: The figure reports the drought index, but this index hasn't been introduced in the text. Please include details about it.

We have added a brief description of the drought index in the Introduction section where Figure 1, which shows the time series of basin-wide drought index, is introduced. (Lines 96-98)

Table 1: I kindly suggest presenting values in percentage points, such as SW and GW, for ease of reading.

We have added percentages in parentheses next to the reported volumes in Table 1 so the reader can easily see the relative amount/importance of water use. We chose to keep the volumetric data as well because we think they are important for understanding regional differences in sectoral water use and surface water vs groundwater use.

Line 100: The Palmer Drought Severity Index is mentioned without an introduction. Can you please provide more details about this index, such as how it is computed, the threshold values associated with different drought conditions, and references?

We have provided a brief description and reference to the Palmer Drought Severity Index methodology. Lines (101-104)

Point n.5, line 107: Perhaps this was also linked with a low population value, hence the area's vulnerability was much smaller than in recent years. In this context, past drought events may have had a smaller impact. Please expand on this concept by connecting point 5 with n.4 with a couple more lines.

As suggested, we have elaborated on how point n4 is related to point n5. Lines (108-115). We explain that the much larger population, particularly in the lower region, has increased the population potentially impacted by drought impacts and has also led to increased sectoral competition for surface water. However, it is difficult to quantify whether past droughts had smaller impacts because there is scant data for the previous drought of record to make such comparisons so we are hesitant to make the statement that past droughts may have been less impactful.

Figure 4: Please place the legend outside the figure.

We have moved the legend outside the figure.

Please specify which reservoirs you are referring to. Line 260 and following: The topic of reservoir management, particularly in dry conditions, is complex. Therefore, it would be interesting to delve deeper into it. Note that many authors have investigated the issue of reservoir management and its impacts on water availability. For example, see [2,3], which have also been cited later in the Discussion section.

We should have been clearer about which reservoirs were affected by 1) record low inflows and 2) large releases for agriculture. We have clarified that "A specific feature of the 2011 to 2015 period that caused severe reservoir drought to persist in the lower region was the absence of any large storm events to replenish storage" (Lines 263-265) and we added "Our analysis showed that extensive irrigation helped partially offset the agricultural impacts in the Colorado Basin, TX However, as shown by Figure 11 management decisions for one sector can reduce or increase impacts to other sectors or even the same sector in another location. An example of cross-sector impacts in the Colorado Basin was agricultural demand in the lower region hastening reservoir drought, which produced cascading impacts to municipal supply availability (triggering conservation measures) and reduced water availability for environmental flows." (Lines 687 - 692).

It's interesting to explore the management strategies for mitigation. For instance, quantifying how the supply strategy mitigated the drought effects would be interesting

We agree that reservoir management is a critical part of drought mitigation, as highlighted throughout the text. However, quantifying reservoirs mitigated impacts would require a counterfactual scenario without reservoir management. A more detailed analysis to quantify the effect of reservoir releases on mitigating drought impacts would require an assessment of

shortages under natural flow conditions which is beyond the scope of this high-level review study.

Line 393: There are two consecutive brackets in this line. I suggest checking the reference style.

We have thoroughly checked every in-text reference for accuracy and consistency.

Line 440: Can you please expand a bit more on this concept, maybe by providing a percentage? For example, "However, agricultural impacts would have been far more severe if losses weren't partially offset by federal assistance and crop insurance (TWDB, 2022b)."

Unfortunately, insurance payments are not available at the county level or water management region. At a state level, the worst drought year in 2011 had over $13 billion in estimated crop losses. We added: "However, agricultural impacts would have been more severe if losses were not partially offset by federal assistance and crop insurance (TWDB, 2022b). For example, at the state level there were $2.6 billion in insurance payments (Collins and Bulut, 2011), while state-level losses were estimated at $13 billion (Anderson et al., 2012). However, the losses reported by Anderson et al. (2012) are gross revenue so the $2.6 billion likely made up for a large fraction of lost profit." (Lines 412-416)

Line 651: " Notable changes to water management and planning include updating policies to conserve water more aggressively during future droughts, new laws to improve regional water planning, and modelling advancements to improve water management and planning." Please consider revising this sentence for better connection to the content.

We agree that this sentence didn't add any significant insight to the content of this Section. We have removed this sentence and instead start Section 4.3 with the first full paragraph.

Paragraph "4.3 Water Management Responses and Planning Innovations" is an interesting topic, but the content should be presented more coherently to avoid a mere "list of information" format.

As stated above, our interpretation of this comment is that it is referring to wanting a more condensed synthesis of findings. However, for the amount of information that is presented in Section 4.3, the text in each of the three main paragraphs is already quite succinct in describing how 1) reservoir management changed (paragraph 1), 2) what modeling improvements were made (paragraph 2), and 3) how new laws were passed to improve drought preparedness (paragraph 3). We worked to condense the content of 4.3 as much as possible.

References:

[1]     Kreibich, H., et al., "The challenge of unprecedented floods and droughts in risk management", Nature. **608**, 80–86 (2022).

[2]     Di Baldassarre, G., Wanders, N., AghaKouchak, A., Kuil, L., Rangecroft, S., Veldkamp, T.I.E., Garcia, M., van Oel, P.R., Breinl, K., Van Loon, A.F., "Water shortages worsened by reservoir effects", Nat. Sustain. **1**, 617–622 (2018).

[3]      Van Loon, A.F., Gleeson, T., Clark, J., Van Dijk, A.I.J.M., Stahl, K., Hannaford, J., Di Baldassarre, G., Teuling, A.J., Tallaksen, L.M., Uijlenhoet, R., Hannah, D.M., Sheffield, J., Svoboda, M., Verbeiren, B., Wagener, T., Rangecroft, S., Wanders, N., Van Lanen, H.A.J., "Drought in the Anthropocene", Nat. Geosci. **9**, 89–91 (2016).

---

## Author Response (AR2)

***Reviewer 2:***

***General Comment***: *The paper now consistently improved after the review. However, there are still some issues that should be addressed before submission.*

We are grateful for the constructive peer review from Reviewer 2 and have revised our draft manuscript to address most of the reviewer's comments. Please find below a detailed account of our responses to the review comments *(in Italics)* below.

***Comment 1***: *Pag.4: the list of key factors that made the drought as a record drought is interesting, however the language is confidential, rather than scientific and the factors are presented in a non-homogeneous style.*

**Response**: We've updated section 1.2 to address the comment. The key factors are now presented in a consistent style.

***Comment 2***: *p.18 "reservoir drought": I kindly suggest changing this way of saying here and elsewhere in the paper. In my opinion, "reservoir drought" is a definition that has many drawbacks. For instance, the reservoir is a structure that may exacerbate or attenuate drought, depending on the management measures. It cannot be associated to an extreme event itself, rather than to a management. Thus, reservoir drought refers maybe to the small inflow that causes low water levels, however, this is not its scientific definition, here we talk about hydrological drought.*

**Response**: We agree and now clarify in the Introduction (Line 43) that reservoir drought can be considered a subclass of Hydrological Drought, which broadly encompasses negative anomalies in surface and subsurface water, such as below-normal groundwater levels or water levels in lakes or decreased river discharge (Van Loon, 2015). Yet, it is also true that "There is no single quantitative definition of drought and drought can be defined by many metrics of water deficit (Kuwayama et al., 2018)". As you nicely articulate, reservoir drought has distinctive causes and effects compared to other forms of drought. As further detailed in our paper, reservoir drought is significantly affected by management decisions and has intricate linkage with meteorological, soil moisture (agricultural), and streamflow drought. The combined effect of increased demand and reduced supply for reservoirs can lead to reduced storage for prolonged periods of time – "reservoir drought" - that has cascading impacts across multiple sectors. Hence, we feel it is important to discuss reservoir drought as a distinct type of drought.

Van Loon, A. F.: Hydrological drought explained, Wires Water, 2, 359-392, 10.1002/wat2.1085, 2015.

Kuwayama, Y., Thompson, A., Bernknopf, R., Zaitchik, B., and Vail, P.: Estimating the Impact of Drought on Agriculture Using the Us Drought Monitor, Am J Agr Econ, 101, 193-210, 10.1093/ajae/aay037, 2019.

***Comment 3***: *Line 484: "To contextualize how unprecedented 2011 inflows were, the lowest inflows during the 1950's drought were approximately four times greater than in 2011", this sentence may be good for a local that knows how extreme the event in the 1950s was, to the others, it is not meaningful. Please consider to rephrase.*

**Response**: We added inflow volumes in million $m^3$/year to the text "...the lowest inflows during the 1950's drought (619 million $m^3$) were approximately four times greater than in 2011 (157 million $m^3$) (Austin Water, 2018)." Also, the following sentence provides the reduction of inflows in terms of average

annual values between 1942 and 2017: "In 2011, inflows to lower region reservoirs were the lowest on record, and only 10.6% of average annual inflows from 1942 to 2017 (Austin Water, 2018)."

**Comment 4**: *Figure 2. Reservoir drought is actually a consequence of hydrological drought, so it is not accurate to place it between the other type of droughts. Also, reservoirs may have different roles during drought events as they are structural measures that may either mitigate or exacerbate drought. Drought can be observed at the reservoir scale as the inflow may reduce and this leads to lower water level, however this may be seen as an impact of drought rather than the drought itself*.

**Response**: Regarding "reservoir drought," please see our response to Comment 2. We modified Figure 2 by changing "Hydrological Drought" to "Streamflow Drought," which makes "Reservoir Drought" a downstream impact of "Streamflow Drought." We also added a connection from "Soil Moisture Drought" to "Reservoir Drought" to indicate that increased demand (for example from Irrigation) can contribute to "Reservoir Drought" in addition to reduced surface flows ("Streamflow Drought"). We have revised the text describing Figure 2 accordingly.

**Comment 5**: *Figure 4. It would be interesting to know if the change in groundwater use is due to a lower availability of groundwater itself. Indeed, it is interesting to see that during drought its use was high, while decreased after the drought period. this may suggest either an attempt to recover the aquifer or a low water availability in the aquifer or a management that preferred to use surface water once it was more available after the drought.*

**Response**: All the potential factors you list for the groundwater use behavior are possibilities. However, the underlying data and documents do not provide information about any specific coordinated efforts to reduce GW use after the drought. The preference in the basin, both from a management and economic consideration is to use SW as it is often much less expensive than GW. We have added the following statements to Section 3.1.1 (Figure 4 interpretation):

Lines 213-217: "Only upper region groundwater use declined in the post-drought period compared to the pre-drought period. In all three regions, post-drought agricultural groundwater use declined compared to the drought period (Figure 4a). The post-drought decline in agricultural groundwater use across the basin could be due to a combination of reduced irrigation demand due to the cessation of meteorological drought, more efficient irrigation technology/practices, and in the middle and lower regions also be influenced by a preference for lower cost surface water when available."

And regarding municipal groundwater use: Lines 227-231: "A consistent pattern in municipal groundwater use shared by all three regions was increased use during the drought followed by reduced use after the drought. This suggests a temporary shift towards groundwater to compensate for reduced surface water supply. The reduction groundwater use following the drought could also be due successful long-term demand management efforts implemented in response to the drought (3.2) and also a preference to use lower cost surface water when available."

**Comment 6:** *Sect. 3.2 is a very interesting section providing further details on drought period. However, it is very long and subdivided into subsections. I would recommend shortening the text taking advantage of the figures and tables presented in the text. The section would highly benefit from discussing findings and data rather than being a list of data.*

**Response:** Much of the extensive revisions in response to the first round of reviews were targeted at condensing the text to only highlight key findings and features of the Figures/Tables. We respectfully disagree with the characterization that the revised text in Section 3.2 is a list of data. While some values are reported, Section 3.2 synthesizes high-level insights from our analysis and adds additional background information on water management and planning responses that are not apparent from inspecting the Figures/Table. In this regard, the text is *complementary* to the Figures/Table rather than a written summary or list of data. We do not think Section 3.2 can be shortened in any substantive way without removing key information and therefore no change has been made to this section.

**Comment 7:** *Figure 11 is very relevant for understanding the processes. However, some impacts may be synthetized in a unique box, while others should be at the same level. For instance, reduced reservoir flow, reduced fresh flow and reduced basin outlet flow can be synthetized as unique box as they are all reduction in river flow, then, the impacts may be different but all stemming from the same box. Again, I'd suggest renaming "reservoir drought". Please also consider changing this figure into a casual loop in which the reduction or increase of a variable is represented by a symbol - or +, respectively. It is better than adding "reduced" or "increased" in each box. Please, also consider subdividing variables into drivers and impacts as it would be easier being read.*

**Response**: Regarding the suggestion of grouping impacts, we are concerned that doing so could omit important granularity of state variables that are important for understanding what specific states contributed to drought impact propagation. Our preference is to retain the current, more detailed, presentation with no change made.

Regarding "reservoir drought," please see our response to Comment 2.

We have implemented the symbology recommendation. Reduction or increase of a variable is now represented by a symbol - or +, respectively, and each link is now annotated. The descriptive words "reduced" or "increased" have been removed from the state labels.

Subdividing drivers and impacts: In Figure 11, Drivers are not colored or labeled, and Impacts are. We now clarify this in the Figure 11 caption.

**Comment 8:** *The Discussion section reports a discussion on a limited amount of aspects linked to drought. It would be interesting to expand the discussion. For instance, the depletion of the aquifer could be discussed, there could be a further discussion on the change in management strategies to build a more resilient environment for possible future drought events.*

**Response**: Added Section 4.2 "Building a More Resilient and Sustainable Water Supply" to the Discussion.

**Comment 9**: *The Conclusions section is missing the key finding of this work. Moreover, in the Conclusions, the diagram is presented as a novel aspect, however it is simply a tool to show relationships between variable/impacts/drivers.*

**Response:** Due to the nuanced and complex nature of the drought impacts and responses there is not a single key finding to highlight. Instead, several key findings are highlighted in the Conclusion, such as:

(1) "Water supply infrastructure (reservoirs, pipelines, canals, and wells) and temporary demand management responses were key for averting severe shortages to non-agricultural sectors."

(2) "Our evaluation of regional water management plans revealed that the drought substantively affected water management planning with large increases in the variety of water supply strategies (supply diversification) and planned municipal supply volume."

(3) "There is no "silver bullet" water management solution for the basin like building a large new reservoir. Instead, a mosaic of supply and demand management strategies are needed to achieve long-term water security."

(4) "Evidence of proactive changes to water management and planning following the drought of record includes the development of more sophisticated water supply planning models, the enactment of more conservative drought management policies, and the passing of several new laws that regulate water planning."

We do not claim that our use of an influence diagram is novel. Regarding the influence diagram, the Conclusions state: "We demonstrate the use of an influence diagram as an effective tool for summarizing cascading regional multisectoral impacts and interactions. Insight into the connectivity between impacts can support adaptive planning and help reduce the vulnerability of negative cascades in other regions (Lawrence et al., 2020)."

**Comment 10**: Regarding the title, the paper does not present a new method to assess drought and discuss impacts and adaptation strategies. I am not sure whether the structure proposed is relevant for other studies. It is important that the authors show why this paper is relevant to the scientific community, and which are the lessons learnt, otherwise it may look like a list of impacts, while it is more than that.

**Response**: Title modified to: "Multisectoral analysis of drought impacts and management responses to the 2008-2015 record drought in the Colorado Basin, Texas." We have revised the following statement in the Conclusions accordingly: " We feel this study offers a "blueprint" that can be followed by future regional drought analyses…."

**Comment 11**: *Please, also consider an extensive English language review.*

**Response**: The paper has been reviewed by a technical English writing specialist at our laboratory. Additionally, no concerns have been raised by Reviewer 1 or the Editor regarding the need for an English language review.

---

## Author Response (AR3)

Editor: "Many thanks for your responses to the reviewer. I have reviewed these and satisfied that you have mostly addressed these comments. I would like to request one minor revision before publication.

I agree with Reviewer 2 that the use of the term 'reservoir drought' could cause confusion for readers (as it is not particularly common to define 'reservoir droughts'). Can you please expand what you mean by reservoir droughts (perhaps on page 2 - Line 45 where you first introduce the term) with some of the text you included in your response (i.e., it is a subclass of hydrological drought and reservoir droughts are significantly impacted by management decisions). This would help to clear the confusion, even if readers do not agree with the terminology."

We have added three sentences to the Introduction (Lines 44-49) describing reservoir drought in more detail:

"Reservoir drought has not been widely studied in the literature (Shah et al., 2024). Shah et al. 2024 define reservoir-based hydrological drought (i.e., reservoir drought) as a period when reservoir storage has persistent negative anomalies due to diminished inflow (streamflow drought), increased net evaporation (meteorological drought), and/or water resource management decisions (i.e., storage releases). Because of the importance of reservoirs to water supply resilience (Kuria and Vogel, 2014), irrigation for food production (Biemans et al., 2011), hydropower, and stream flow (Wanders and Wada, 2015), reservoir droughts can have significant socioeconomic, energy, and environmental implications."